# FlowDrag: 3D-aware Drag-based Image Editing with Mesh-guided Deformation Vector Flow Fields

**Gwanhyeong Koo** [1]  **Sunjae Yoon** [1]  **Younghwan Lee** [1]  **Ji Woo Hong** [1]  **Chang D. Yoo** [1]

## Abstract

Drag-based editing allows precise object manipulation through point-based control, offering user convenience. However, current methods often suffer from a geometric inconsistency problem by focusing exclusively on matching user-defined points, neglecting the broader geometry and leading to artifacts or unstable edits. We propose **FlowDrag**, which leverages geometric information for more accurate and coherent transformations. Our approach constructs a 3D mesh from the image, using an energy function to guide mesh deformation based on user-defined drag points. The resulting mesh displacements are projected into 2D and incorporated into a UNet denoising process, enabling precise handle-to-target point alignment while preserving structural integrity. Additionally, existing drag-editing benchmarks provide no ground truth, making it difficult to assess how accurately the edits match the intended transformations. To address this, we present VFD (VidFrameDrag) benchmark dataset, which provides ground-truth frames using consecutive shots in a video dataset. FlowDrag outperforms existing drag-based editing methods on both VFD Bench and DragBench.

## 1. Introduction

The advancements in text-to-image (T2I) generation diffusion models (Ramesh et al., 2022; Saharia et al., 2022; Nichol et al., 2021; Li et al., 2023) have significantly enhanced image generation capabilities. Leveraging pretrained T2I diffusion models (Rombach et al., 2022) trained on large scale dataset, the image editing field has also seen substantial progress. Text-based image editing (Hertz et al., 2022;

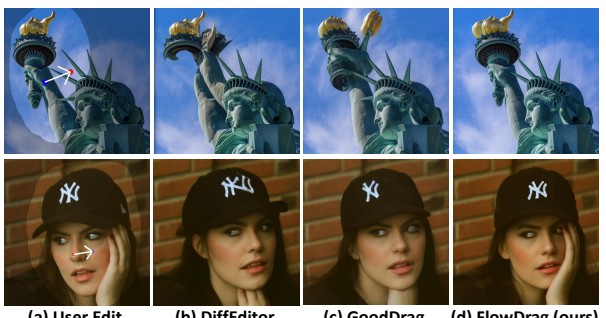

*Figure 1.* **Results of drag-based image editing.** While other methods only optimize around user-specified points, failing to preserve the Statue of Liberty's structure (first row), FlowDrag maintains overall integrity. In the second row, FlowDrag stably rotates the woman's face from her nose without distorting her hat or hand, whereas other methods fail to maintain geometric consistency.

Tumanyan et al., 2023; Cao et al., 2023; Koo et al., 2024a), which uses user-provided prompts, has shown promising results but often struggles with precise and fine-grained edits. Even slight variations in text can lead to vastly different results, making it difficult for users to achieve consistent and detailed modifications, thereby highlighting the challenges of achieving precision in text-to-image editing. To address this, DragGAN (Pan et al., 2023) introduced a point-dragging method, enhancing user convenience for more precise editing. However, due to the constraints of GANs, struggled with general image performance. This limitation led to interest in exploring drag-based image editing using diffusion models, which can be categorized into two main approaches: motion-based (Shi et al., 2024; Ling et al., 2023; Liu et al., 2024; Zhang et al., 2024) and gradient-guidance-based (Mou et al., 2023; 2024) methods. The motion-based approach in drag-based image editing consists of two processes: motion supervision and point tracking. Motion supervision measures the difference between the handle and target points in the UNet decoder's feature map and applies it to optimize the latent, gradually shifting the handle point toward the target. Subsequently, point tracking updates the handle point's position in the feature map, ensuring alignment with the progressively edited result. In contrast, gradient-based methods derive inspiration from score-based (Song et al., 2020b; Dhariwal & Nichol, 2021)

---

[1]Korea Advanced Institute of Science and Technology (KAIST). Correspondence to: Chang D. Yoo <cdyoo@kaist.ac.kr>.

*Proceedings of the 42$^{nd}$ International Conference on Machine Learning*, Vancouver, Canada. PMLR 267, 2025. Copyright 2025 by the author(s).

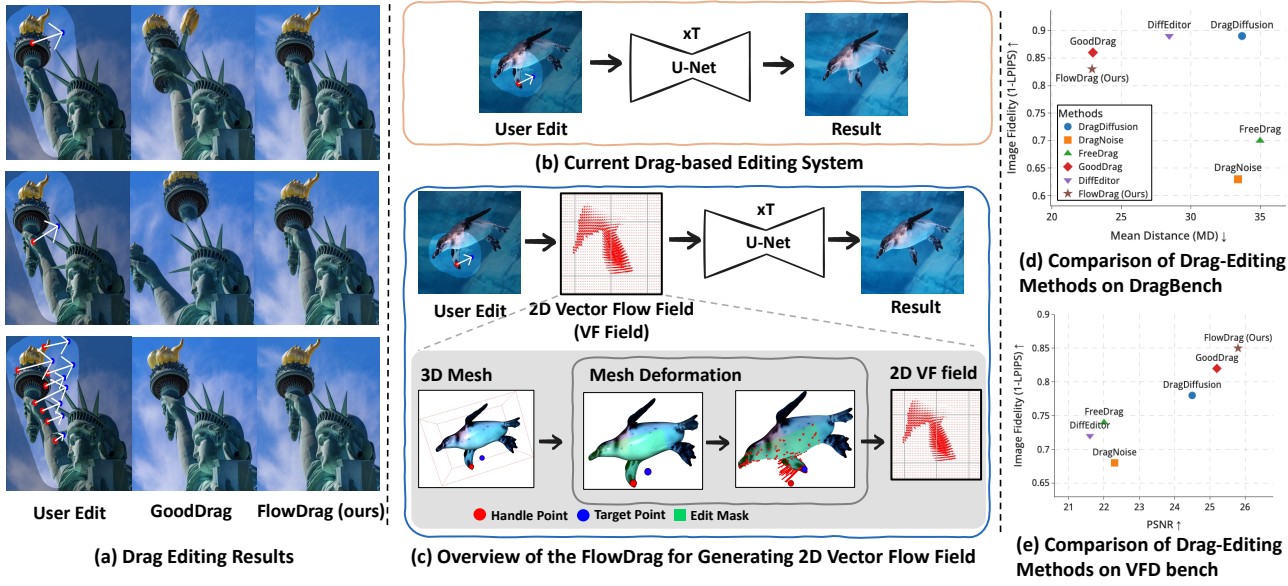

*Figure 2.* (a) GoodDrag vs. FlowDrag (ours). GoodDrag fails to preserve object geometry, while FlowDrag retains structural integrity. (b) Existing drag-editing systems focus only on moving user-defined handle points to target positions. (c) FlowDrag overview: from 3D mesh creation to generating a 2D vector flow field. (d) Comparison of FlowDrag with other methods on DragBench. (e) Comparison of FlowDrag with other methods on our VFD-Bench dataset.

diffusion model, utilizing gradient guidance governed by an energy function to perform the drag-based editing.

However, existing drag editing methods often suffer from a geometric inconsistency problem, leading to artifacts or significant deviations from the original object's structure. These issues become evident in the first row of Fig.1(b)-(c), where the existing model fails to preserve the statue's integrity, causing the arm and torch to become noticeably altered and leading to visible artifacts. Similarly, in the second row of Fig.1, we attempt to rotate a woman's face by placing a drag arrow on her nose. While her hat and the hand resting on her face should naturally follow this rotation, existing methods shift the nose but fail to preserve the hat and hand, resulting in unnatural deformation. We identified that these problems primarily stem from current methods' exclusive focus on matching feature correspondence between user-defined handle and target points, neglecting the broader geometric context of the image, as shown in Fig. 2(b). This issue is particularly prominent in edits that require preserving rigid parts of the object. Therefore, we define any transformation that must maintain rigidity (such as rotation, relocation, or pose changes) as a "Rigid Edit." In contrast, transformations that do not require strict rigidity, such as rescaling (which alters proportions), are referred to as "Non-rigid Edits." In this paper, we focus on "Rigid Edits" to ensure structural integrity during drag editing.

To overcome these limitations, we propose **FlowDrag**, a novel method designed to ensure stable and accurate image edits by preserving geometric information. Our approach

proceeds through several stages. First, we construct a 3D mesh from the original image to represent the object's geometry. Second, we employ a progressive SR-ARAP (Levi & Gotsman, 2014) approach to deform the mesh from user-defined handle points to target points, preserving the object's geometric integrity throughout the transformation, as illustrated in Fig. 2(c). Third, after the mesh deformation, we compute the differential coordinates between the original and modified mesh and project them onto a 2D vector field. Finally, this vector field is integrated into the motion supervision phase of the UNet's denoising process, which enhances both spatial accuracy and edit stability. Unlike existing methods restricted to user-defined drag editing points, FlowDrag leverages a continuous displacement field derived from mesh deformations, preserving both spatial and geometric coherence for more reliable results.

We also propose a new drag-editing benchmark called **VFD-Bench**. Existing benchmarks, such as DragBench, do not provide ground-truth edited images for each input, making it difficult to accurately assess editing quality. The Image Fidelity metric (1-LPIPS) measures similarity between the original input and the edited result, often assigning lower scores to successful geometry-preserving edits. For instance, Fig.1(d) preserves geometry well but receives a relatively low 1-LPIPS score of 0.72 compared to Fig.1(a), indicating that this metric does not fully capture structural accuracy. To address this, VFD-Bench provides video-derived datasets where consecutive frames serve as paired input images and ground-truth edits, enabling more precise evaluation. Our FlowDrag achieves the highest MD metric on DragBench,

and as shown in Fig.2(d), also outperforms competing approaches on the new VFD-Bench, validating its effectiveness in maintaining geometric consistency.

## 2. Related Work

### 2.1. Text-based Image Editing

Early GAN-based methods (Patashnik et al., 2021; Xia et al., 2021) edited images by inverting them into StyleGAN latent spaces conditioned on textual descriptions. However, these methods had limited flexibility due to inherent trade-offs between generalized editing capability and reconstruction quality. Recently, diffusion-based methods (Kawar et al., 2023; Couairon et al., 2022; Brooks et al., 2023) enabled more precise and diverse edits by directly guiding the diffusion process with text prompts. Prompt-to-Prompt (Hertz et al., 2022) and Plug-and-Play (Tumanyan et al., 2023) further improved editing precision through attention feature injection mechanisms. More recent work also explored adjusting object poses and perspectives (Cao et al., 2023; Koo et al., 2024a; Yoon et al., 2024a). With advances in text-based editing techniques, various studies utilized these methods for tasks such as improving editing efficiency (Koo et al., 2024b; Deutch et al., 2024), human image animation (Yoon et al., 2024c), and virtual try-on (Hong et al., 2025). However, text-based methods often lack the precision required for fine-grained control, as minor textual variations can result in unintended edits. This limitation has motivated the emergence of drag-based editing methods, which offer more direct and intuitive control.

### 2.2. Drag-based Image Editing

Drag-based image editing modifies images using user-drawn drags, offering precise and interactive control for tasks such as rotation, relocation, and rescaling. While early GAN-based methods (Pan et al., 2023) suffered from limited edit fidelity, diffusion models significantly improved both editing quality and diversity. Diffusion-based drag editing is divided into two categories: motion-based and gradient-guidance-based. Motion-based methods (Shi et al., 2024; Ling et al., 2023; Liu et al., 2024; Zhang et al., 2024) rely on motion supervision and point tracking to iteratively shift handle points toward targets, preserving the original structure. Specifically, DragDiffusion optimizes the DDIM latent at a specific timestep ($t$=35), while Drag Your Noise targets bottleneck features in the U-Net across all timesteps. Good-Drag further introduces the AlDD framework, alternating drag operations and denoising across multiple timesteps to reduce cumulative changes and enhance fidelity. In contrast, gradient-guidance-based methods (e.g., DragonDiffusion (Mou et al., 2023), DiffEditor (Mou et al., 2024)) use gradient updates driven by an energy function derived from score-based diffusion models (Song et al., 2020b; Dhari-

wal & Nichol, 2021), enabling more creative edits but often causing artifacts or reduced fidelity. Although motion-based approaches better preserve visual fidelity, they still struggle with geometric integrity due to limited structural understanding. To address this, we incorporate 3D mesh deformation to add explicit geometric information in 2D.

## 3. Preliminary

### 3.1. DDIM Inversion

DDIM (Song et al., 2020a) eliminates the stochastic elements of DDPM (Ho et al., 2020), producing a deterministic, non-Markovian process for precise control over diffusion steps. A U-Net denoiser network, $\epsilon_\theta$, enables both sampling (Eq. (1)) from noise to image and inversion (Eq. (2)) from image to noise. Here, $\alpha_t$ denotes the noise schedule at step $t$, and $z_t$ is the latent representation at that step.

$$z_{t+1} = \sqrt{\frac{\alpha_{t+1}}{\alpha_t}}\, z_t + \left(\sqrt{1 - \frac{\alpha_{t+1}}{\alpha_t}} - 1\right)\epsilon_\theta(z_t, t), \quad (1)$$

$$z_t^* = \sqrt{\frac{\alpha_t}{\alpha_{t-1}}}\, z_{t-1}^* + \left(\sqrt{\frac{1-\alpha_t}{\alpha_{t-1}}} - 1\right)\epsilon_\theta(z_{t-1}^*, t-1). \quad (2)$$

In drag editing, DDIM Inversion is applied to obtain the final DDIM latent $z_t$ by progressively adding noise from $z_0$ to $z_t$. As detailed in Section 3.2, this latent is optimized via motion supervision to refine editing results.

### 3.2. Diffusion Latent Optimization in Drag Editing

**Motion Supervision**   Motion-based drag editing typically applies a motion supervision loss, $\mathcal{L}_{\mathrm{ms}}$, to iteratively shift $n$ handle points $\{h_i^k\}_{i=1}^n$ toward target points $\{t_i\}_{i=1}^n$. Formally, the motion supervision loss is defined as:

$$\mathcal{L}_{\mathrm{ms}}(z_t^k) = \sum_{i=1}^{n} \sum_{q \in \mathcal{P}(h_i^k, R)} \left\| F_{q+\delta_i}(z_t^k) - \mathrm{sg}(F_q(z_t^k)) \right\|_1$$
$$+ \lambda \left\| \left(z_{t-1}^k - \mathrm{sg}(z_{t-1}^0)\right) \odot \left(1 - M\right) \right\|_1, \quad (3)$$

where $\delta_i = \frac{t_i - h_i^k}{\| t_i - h_i^k \|_2}$ is the normalized direction from $h_i^k$ to $t_i$, $\mathcal{P}(h_i^k, R)$ is a patch of radius $R$ around $h_i^k$, $F$ denotes the U-Net feature map, $k$ is denoising timestep and $\mathrm{sg}(\cdot)$ is the stop-gradient operator. The term $\lambda$ weights the regularization that keeps $\{z_{t-1}^k\}$ close to the reference $\{z_{t-1}^0\}$ outside the masked region $M$. At each iteration, we compute $\partial \mathcal{L}_{\mathrm{ms}}/\partial z_t^k$ and update $z_t^k$ via gradient descent:

$$z_t^{k+1} = z_t^k - \eta\, \frac{\partial\, \mathcal{L}_{\mathrm{ms}}(z_t^k)}{\partial\, z_t^k}, \quad (4)$$

where $\eta$ is the learning rate. This process gradually moves each handle point closer to its target in the latent space.

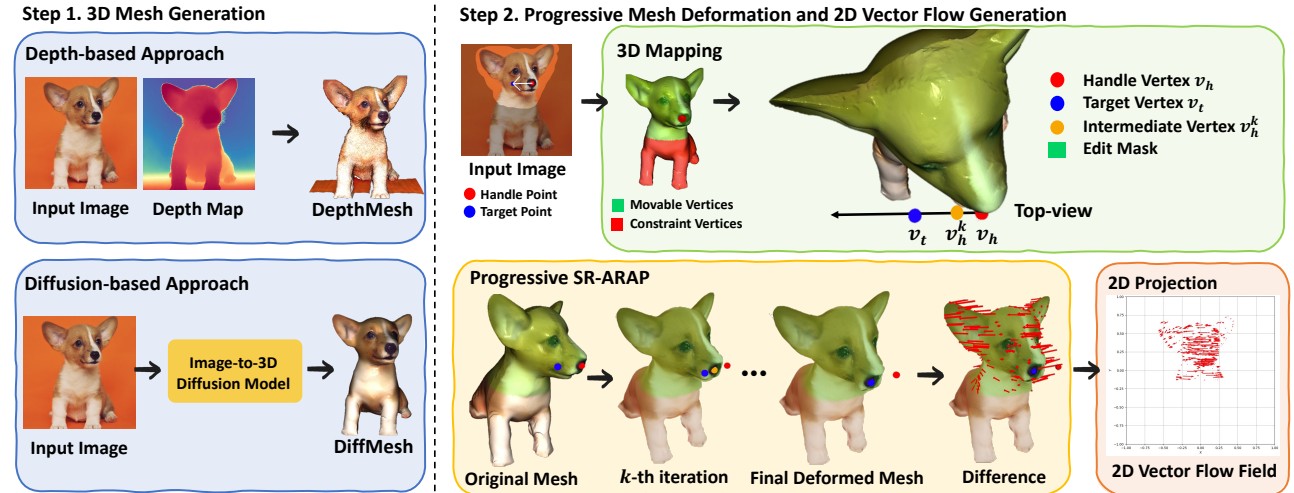

Figure 3. **Overview of the FlowDrag pipeline.** (a) Step 1: 3D mesh generation using depth-based and diffusion-based approaches. (b) Step 2: Progressive mesh deformation via SR-ARAP, with differences projected as a 2D vector flow field.

**Point Tracking**  Following motion supervision, point tracking updates the handle points $\{h_i^k\}$ by searching for the best matching features within a local patch. Specifically,

$$h_i^{k+1} = \underset{q \in \mathcal{P}(h_i^k, R_2)}{\arg\min} \left\| F_q(z_t^{k+1}) - F_{h_i^0}(z_t) \right\|_1, \quad (5)$$

where $\mathcal{P}(h_i^k, R_2)$ is a square patch of radius $R_2$ around $h_i^k$. This step aligns each handle point with the corresponding features in the updated latent. Alternating motion supervision and point tracking guides handle points progressively closer to their targets.

### 3.3. Geometric Mesh Deformation

In this section, we introduce widely utilized geometric mesh deformation methods, the As-Rigid-As-Possible (ARAP) (Sorkine & Alexa, 2007) approach and its enhanced variant, Smoothed Rotation As-Rigid-As-Possible (SR-ARAP) (Levi & Gotsman, 2014). Let $M = (V, F)$ represent the source mesh, where $V$ consists of vertices $v_i = (x_i, y_i, z_i)$ in $\mathbb{R}^{3 \times V}$, and $F$ forms faces that are triangles formed by these vertices. The deformed mesh is denoted as $\hat{M} = (\hat{V}, F)$, where $\hat{V}$ consists of the deformed vertex positions $\hat{v}_i = (\hat{x}_i, \hat{y}_i, \hat{z}_i)$, maintaining the same connectivity.

**ARAP.**  ARAP aims to preserve local rigidity while allowing controlled deformations. Initially, users designate certain vertices as *constraints*, including vertices explicitly moved to desired positions (e.g., handle points moved to target points in drag editing) and vertices outside the editable region that remain unchanged. The remaining vertices are considered *movable*. Setting these constraint vertices transforms the original mesh $V$ to the new constraint positions $\hat{V}$, from which optimization begins. The ARAP method adjusts movable vertex positions by minimizing an energy

function, maintaining local rigidity under fixed constraints. The energy function is defined as:

$$E_{\text{ARAP}}(M) = \sum_{i \in V} \sum_{j \in N(i)} w_{ij} \left\| s_i R_i (\hat{v}_i - \hat{v}_j) - (v_i - v_j) \right\|^2, \quad (6)$$

where $R_i$ and $s_i$ are internally optimized rotation matrices and local scale factors for each vertex $i$ (with $s_i$ set to 1, preventing scale changes). The terms $w_{ij}$ represent cotangent weights, reflecting the stiffness or rigidity between vertices. The notation $N(i)$ denotes the set of neighbors for vertex $i$, specifically those vertices directly connected by an edge, and $j$ denotes each neighboring vertex within this set. Here, $\hat{v}_i, \hat{v}_j$ represent vertex positions optimized during deformation, whereas $v_i, v_j$ refer to positions from the original undeformed mesh. During the optimization process, these positions $\hat{v}_i, \hat{v}_j$ are iteratively updated to minimize the energy function, with only *movable* vertices adjusted, while *constraints* remain fixed as boundary conditions.

**SR-ARAP.**  Smoothed Rotation ARAP extends the ARAP formulation by adding a rotation-consistency term:

$$E_{\text{SR-ARAP}}(M) = E_{\text{ARAP}}(M) + \alpha \sum_{i \in V} \sum_{j \in N(i)} \|R_i - R_j\|^2, \quad (7)$$

where $R_i$ and $R_j$ are the per-vertex rotation matrices, $\alpha$ is a regularization parameter. This extra term penalizes significant rotational discrepancies between adjacent vertices, resulting in smoother deformations. Specifically, a larger rotation difference $\|R_i - R_j\|^2$ increases the energy, prompting the optimization process to minimize these differences. As a result, rotations among adjacent vertices remain as consistent as possible, ensuring smoother and more natural deformations. These formulations facilitate controlled mesh deformation, beneficial for achieving smoother and locally rigid structural modifications.

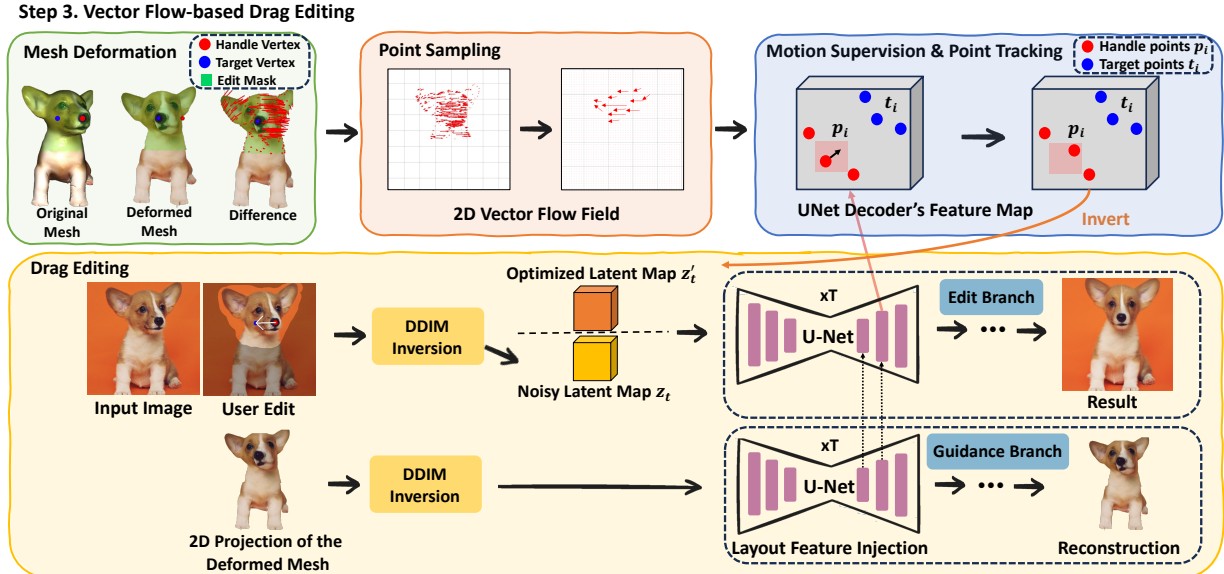

*Figure 4.* **Step 3: Vector Flow-Based Drag Editing in FlowDrag.** The 2D vector flow field is employed for motion supervision and point tracking. Meanwhile, the 3D deformed mesh is projected onto a 2D plane and incorporated as a layout feature through the guidance branch at an early denoising stage. Through these processes, geometric consistency is enhanced during drag editing.

## 4. Method

FlowDrag addresses the geometric inconsistency problem in drag-based editing by incorporating 3D mesh deformation. First, we generate a 3D mesh from the input image using a depth-based or diffusion-based approach (Fig. 3-Step 1). Next, we deform the mesh using progressive SR-ARAP and derive a 2D vector flow from the deformed mesh (Fig. 3-Step 2). Finally, we integrate the 2D vector flow into the motion-based drag-edit pipeline to achieve geometry-aware movements, and utilize the 2D projection of the deformed mesh as layout features to preserve structural consistency and ensure accurate shape preservation (Fig. 4-Step 3). The following sections describe each of these steps in detail.

### 4.1. 3D Mesh Generation: Depth-based and Diffusion-based Approaches

We first generate a 3D mesh from the input image by extracting a depth map and using it as a foundation. For depth extraction, we adopt the Marigold (Ke et al., 2024) model, which reliably provides depth information from a single image within a few seconds. The depth-based mesh construction then proceeds in three stages. First, in the vertex mapping step, each pixel's depth value is converted into a corresponding vertex coordinate in 3D space. Next, the facet formation step connects adjacent vertices whose depth values are sufficiently similar, forming smooth surfaces. Finally, the artifact reduction step excludes connections between vertices with large depth differences, minimizing discontinuities and effectively separating the foreground object from its background. Algorithm 1 in the appendix

summarizes this procedure in detail.

Although this depth-based approach is simple and fast, it cannot account for unseen regions from a single viewpoint. Therefore, we additionally employ image-to-3D diffusion models (Xiang et al., 2024; Zhao et al., 2025), which infer hidden structures to produce more complete meshes. We refer to the mesh generated by our depth-based approach as DepthMesh, and the one produced by the diffusion-based approach as DiffMesh, as illustrated in Fig. 3-Step 1. We employ both meshes in our experiment to explore different levels of geometry detail and completeness.

### 4.2. Progressive Mesh Deformation and 2D Vector Flow Generation

We begin by mapping the user-defined handle point, target point, and mask from the 2D drag-editing setting onto our 3D mesh $M = (V, F)$. In this mapping, a handle vertex $v_h \in V$ corresponds to the handle point, and its target vertex $v_t \in V$ corresponds to the target point. The masked region indicates which vertices are *movable*, while all other vertices become *constraints* (see Fig. 3-Step2). Following the ARAP principle, each movable vertex adjusts its position by minimizing the ARAP energy, preserving local rigidity. In contrast, the constrained vertices remain fixed at their designated positions.

**Progressive Deformation with SR-ARAP.** In ARAP-based methods, the handle vertex $v_h$ is typically moved directly to its target position $v_t$, followed by mesh deformation through ARAP energy minimization. However, when

the distance between $v_h$ and $v_t$ is large, directly moving $v_h$ can cause abrupt local distortions and unnatural deformation (Sorkine & Alexa, 2007; Chen et al., 2017). Such distortions arise because large vertex displacements significantly stretch local edges, increasing ARAP energy and potentially causing convergence to suboptimal local minima. To mitigate these issues, we propose a progressive deformation strategy with two components. First, we incrementally move the handle vertex toward its target over $K$ iterations:

$$v_h^{(k+1)} = v_h^{(k)} + \lambda\big(v_t - v_h^{(k)}\big), \quad 0 < \lambda \leq 1, \quad (8)$$

where $v_h^{(0)}$ is the initial handle position, $v_h^{(K)} = v_t$, and $0 \leq k < K$. The parameter $\lambda$ controls the fraction of remaining distance covered at each step, smoothly distributing large displacements. Second, we introduce an *Inter-Step Smoothness* term to the SR-ARAP energy (Eq. 9), penalizing large vertex displacements between iterations to ensure gradual and stable deformation:

$$E_{\text{SR-ARAP+InterStep}}\big(\hat{M}^{(k+1)}\big) = E_{\text{SR-ARAP}}\big(\hat{M}^{(k+1)}\big) \\ + \beta \sum_{i \in V} \big\|\hat{v}_i^{(k+1)} - \hat{v}_i^{(k)}\big\|^2, \quad (9)$$

where $\hat{v}_i^{(k)}$ and $\hat{v}_i^{(k+1)}$ are the positions of the movable vertices at iterations $k$ and $k+1$. The parameter $\beta$ adjusts the strength of this regularization, with higher values promoting smoother transitions and smaller vertex movements, while lower values allow larger displacements.

**2D Vector Flow Generation.** After $K$ iterations, the original mesh $M$ converges to a final deformed mesh $\hat{M}$. We then project both meshes onto the 2D image plane, denoted by $\pi(M)$ and $\pi(\hat{M})$, respectively. The 2D vector flow $\Phi$ is defined based per-vertex displacements:

$$\Phi = \big\{\big(\Delta x_i, \ \Delta y_i\big) \ \big| \ \Delta x_i = x_i' - x_i, \ \Delta y_i = y_i' - y_i\big\}, \quad (10)$$

where $(x_i, y_i)$ and $(x_i', y_i')$ are the 2D coordinates of vertex $i$ in $\pi(M)$ and $\pi(\hat{M})$. Thus, by capturing how each vertex moves from $M$ to $\hat{M}$, the vector flow $\Phi$ encodes the geometric changes induced by our progressive SR-ARAP approach. In the next section, we show how both $\Phi$ and $\pi(\hat{M})$ integrate into the motion-based drag-edit pipeline, providing geometry-aware guidance for image editing.

### 4.3. Vector Flow-based Drag Editing

**Vector Flow Sampling** We focus on selecting an optimal set of vectors from the 2D flow field $\Phi$ for motion supervision (Eq.3) and point tracking (Eq. 5). First, we uniformly sample an $N \times N$ grid of candidate vectors within the edit mask, taking $N$ positions along each axis. To refine this set, we explore two approaches: (1) Magnitude-based sampling, which sorts all candidates by displacement magnitude

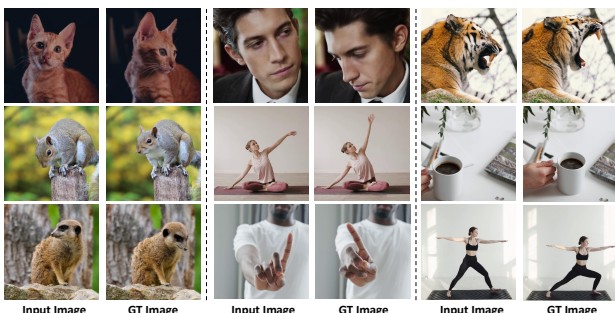

| Input Image | GT Image | Input Image | GT Image | Input Image | GT Image |

*Figure 5.* **Example images from VFD-Bench.** We constructed this drag-based image editing dataset by selecting closely spaced frames from DAVIS, LOVEU-TGVE, and copyright-free Pexels videos, focusing on noticeable changes in pose or structure.

and keeps only the top few, and (2) Uniform sub-sampling, which retains vectors at regular intervals to ensure broad coverage. Either method yields a final subset $\hat{\Phi}$, typically containing 5–30 vectors that capture the most significant or representative displacements. We then restrict the summation in Eq.3 to $\mathbf{q} \in \hat{\Phi}$, focusing the motion supervision on these carefully selected flow vectors.

**Layout Feature Injection** In addition to leveraging our vector flow for latent optimization, we introduce a guidance branch utilizing the 2D projection of the deformed mesh, $\pi(\hat{M})$, to provide complementary geometric context. Specifically, as illustrated in Fig. 4, we first invert $\pi(\hat{M})$ via DDIM Inversion to obtain a latent noise representation. During the denoising process, we inject selected attention features from this representation into the primary drag-edit branch. Previous diffusion studies (Wu et al., 2023b; Yoon et al., 2024b) have shown that earlier timesteps establish broad structural outlines, while later timesteps refine finer details. Following this insight, we inject attention features only from earlier or intermediate timesteps, embedding approximate layout information derived from the deformed mesh. This ensures broader geometric context is transferred without imposing overly specific or potentially conflicting details, as the deformed mesh may not perfectly align with the user's final desired details. Thus, this layout feature injection complements vector-flow-based optimization by explicitly providing structural context, guiding the main edit branch toward a more geometrically consistent edited result.

## 5. VFD-Bench Dataset

Existing drag-based editing benchmarks, such as Drag-Bench, provide input images along with user-defined handle/target points and masks, but do not include ground-truth (GT) edited images. As a result, commonly used metrics like Image Fidelity (IF) and Mean Distance (MD) are computed between the input and the edited output. Specifically, IF (measured as 1-LPIPS) assesses how closely the edited

image resembles the original, while MD uses DIFT features to measure how effectively the handle points have moved to their targets. However, in cases where rotation or pose changes are successfully introduced, IF often yields low scores simply because the edited object differs from the original layout, even though the edit is successful.

To address these limitations, we propose **VFD-Bench**, a new dataset that provides an explicit GT image for each input. As illustrated in Fig. 5, VFD-Bench is constructed by selecting pairs of video frames (from sources such as DAVIS (Pont-Tuset et al., 2017), LOVEU-TGVE (Wu et al., 2023a), TVR (Lei et al., 2020) and copyright-free clips on Pexels[1]) where an object undergoes a clear change in pose or structure. In total, we construct 250 input-GT pairs suitable for drag-based editing. For each pair, the handle points and target points are defined based on the differences between the input and ground-truth (GT) images, with 1–5 drags assigned per sample. Further details regarding the dataset annotation process and the number of images per category are provided in Appendix C.

### 5.1. Evaluation Metrics

Unlike previous benchmarks, VFD-Bench enables direct comparison of edited images to actual GT results. We measure Image Fidelity using both PSNR (at the RGB level) and LPIPS (at the feature level). For assessing how well handle points align with target points, we retain the Mean Distance (MD) metric. Since video frames can contain changing backgrounds unrelated to the target object, we compute all metrics within the user mask to focus on the edited region. This approach allows for a more reliable and detailed evaluation of drag-based editing methods.

## 6. Experiments

### 6.1. Implementation Details

We validate FlowDrag using the pre-trained Stable Diffusion 1.5 model (Rombach et al., 2022), processing images at a resolution of $512 \times 512$. For image encoding and decoding, we use VQ-VAE (Razavi et al., 2019). Following prior motion-based approaches (Ling et al., 2023; Zhang et al., 2024; Liu et al., 2024), we fine-tune the input image with LoRA (Hu et al., 2021) (rank = 16) for 200 steps. DDIM Inversion is applied up to step 38 (75% of the total 50 denoising steps), as in (Zhang et al., 2024), with layout feature injection at timestep $t' = 30$. For 3D mesh generation, we primarily utilize DiffMesh. However, when DiffMesh exhibits significant artifacts or deviates substantially from the original image, we employ DepthMesh instead. In Progressive SR-ARAP mesh deformation, we compute $w_{ij}$ in

---

[1]Pexels: copyright-free videos at (https://www.pexels.com/)

---

*Table 1.* Performance of recent drag-based editing methods on the DragBench dataset.

| Method | 1−LPIPS↑ | MD↓ |
|--------|----------|-----|
| **DragBench** | | |
| DiffEditor (Mou et al., 2024) | **0.89** | 28.46 |
| DragDiffusion (Shi et al., 2024) | **0.89** | 33.70 |
| DragNoise (Liu et al., 2024) | 0.63 | 33.41 |
| FreeDrag (Ling et al., 2023) | 0.70 | 35.00 |
| GoodDrag (Zhang et al., 2024) | 0.86 | 22.96 |
| **FlowDrag (ours)** | 0.82 | **22.88** |

*Table 2.* Performance of recent drag-based editing methods on the VFD-Bench dataset.

| Method | PSNR↑ | 1−LPIPS↑ | MD↓ |
|--------|-------|----------|-----|
| **VFD-Bench** | | | |
| DiffEditor (Mou et al., 2024) | 21.6 | 0.72 | 24.88 |
| DragDiffusion (Shi et al., 2024) | 24.5 | 0.78 | 36.52 |
| DragNoise (Liu et al., 2024) | 22.3 | 0.68 | 36.21 |
| FreeDrag (Ling et al., 2023) | 22.0 | 0.74 | 38.32 |
| GoodDrag (Zhang et al., 2024) | 25.2 | 0.82 | 25.65 |
| **FlowDrag (Ours)** | **26.3** | **0.85** | **24.51** |

Eq. 6 using Open3D library, and set $\alpha$ between 0.2 and 0.4. Additionally, we conduct an ablation study on the Inter-Step Smoothness term (Section 6.4), exploring $\beta \in [0, 1]$ in Eq. 9. To integrate vector flow into drag editing, we uniformly sample points from a $20 \times 20$ grid ($N = 20$) within the 2D flow field over the edit mask, selecting between 5 and 30 points for optimization. This setup balances computational efficiency with adequate coverage of the flow field.

### 6.2. Datasets and Evaluation Metrics

**Datasets** We evaluate FlowDrag on the DragBench (Shi et al., 2024) and our proposed VFD-Bench. DragBench contains 205 images of diverse content, along with 349 pairs of handle and target points. Each image has one or more dragging instructions (i.e., handle–target point pairs) and a mask that specifies the editable region. VFD-Bench, introduced in Section 5, includes 250 images selected from video sources (DAVIS, TGVE, TVR, and Pexels), each paired with a ground-truth (GT) edit.

**Metrics** We evaluate both fidelity and precision for each edited image. On DragBench, where no ground truth (GT) is provided, we measure Image Fidelity (IF) as 1-LPIPS and compute the Mean Distance (MD) using DIFT (Tang et al., 2023) to track how well handle points move to their targets. In VFD-Bench, which provides GT images, we quantify fidelity with both PSNR (RGB-level) and LPIPS (feature-level) while retaining MD for point alignment, as described in Section 5. Since VFD-Bench frames can involve irrelevant background changes, we compute metrics within the user-defined mask to focus on the edited region.

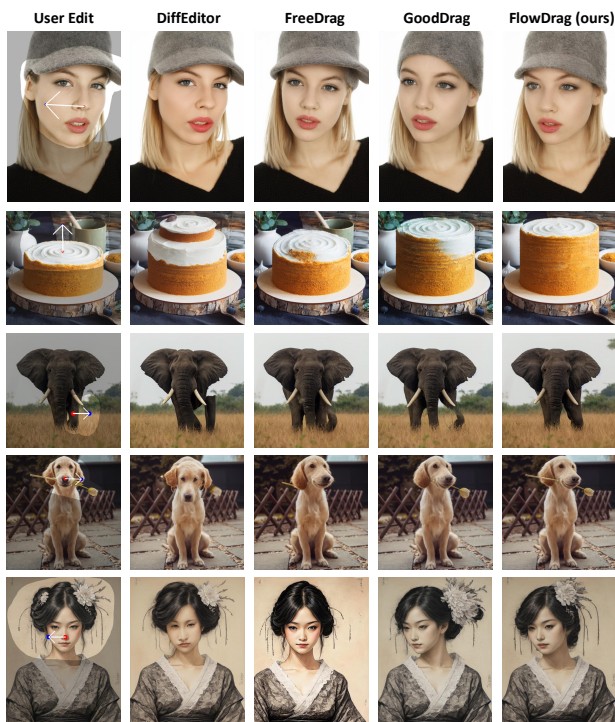

*Figure 6.* **Qualitative results with recent drag-based image editing systems.** Our FlowDrag produces outputs that maintain geometric consistency more effectively than other methods.

### 6.3. Experimental Results

**Qualitative Comparisons.** We compare FlowDrag with current drag-based editing methods in Fig. 6. Although motion-based approaches generally outperform the gradient-guided DiffEditor, FreeDrag and GoodDrag often neglect the object's broader spatial structure by relying solely on user drag points. In contrast, FlowDrag leverages a vector flow field from mesh deformation, enabling more cohesive transformations. For example, in the first row of Fig. 6, only one drag point is provided. DiffEditor fails to move the face, FreeDrag relocates the face but leaves the hat unchanged, and GoodDrag removes the brim. However, FlowDrag adjusts both the face and the hat, illustrating how spatially informed vectors yield more natural edits with fewer artifacts. Additional qualitative comparisons on DragBench and VFD-Bench are provided in Appendix F.

**Quantitative Results.** On DragBench, FlowDrag achieves the best Mean Distance (MD), indicating effective dragging, as shown in Table 1. However, DiffEditor scores highest on the 1-LPIPS fidelity metric because it induces minimal edits. In VFD-Bench, which includes actual ground-truth images from real video frames, FlowDrag outperforms all methods in PSNR, 1-LPIPS, and MD, demonstrating its effectiveness in preserving geometric consistency while delivering accurate edits.

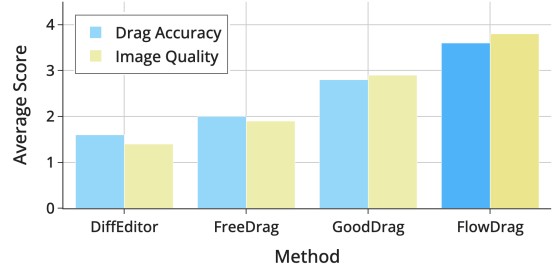

*Figure 7.* **User study on drag accuracy and image quality.**

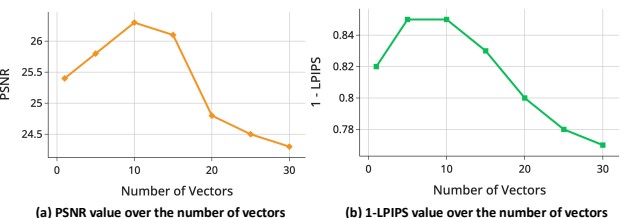

*Figure 8.* **Impact of vector count on (a) PSNR and (b) 1-LPIPS.** When 10 vectors are sampled, both metrics achieve their highest values, indicating the importance of optimal vector selection.

**User Study.** We conducted a user study on 50 images from DragBench and VFD-Bench, comparing FlowDrag with DiffEditor, FreeDrag, and GoodDrag. We recruited 25 volunteers to rank each method's edited results (4 = best, 1 = worst) based on drag accuracy and image quality. As shown in Fig. 7, FlowDrag consistently received higher scores than the other methods in both aspects.

### 6.4. Ablation Study

We conducted an ablation study in FlowDrag on VFD-Bench by evaluating the following three aspects: (1) influence of the regularization parameter $\beta$ in progressive SR-ARAP deformation, (2) effect of selected vector count on performance, (3) magnitude-based vs. uniform sub-sampling in 2D vector flow field sampling.

**Influence of $\beta$ in Progressive SR-ARAP** We investigate the effect of varying the parameter $\beta$ of the Inter-Step Smoothness term in Eq.9 on local rigidity preservation during mesh deformation. To quantify this, we introduce a *rigidity measure* (see Appendix A.1) based on mean edge length ratios (MELR) and the Mean ARAP Error ($\mathrm{mARAP_{Error}}$) between the original mesh $M$ and its deformed counterpart $\hat{M}$. The results for $\beta \in \{0.2, 0.4, 0.6, 0.8, 1.0\}$ are presented in Table 4. Higher $\beta$ values penalize large vertex displacements between iterations, resulting in smoother deformations but potentially restricting flexibility. Notably, $\beta = 0.8$ yields the best results, achieving maximal mean edge length ratio and mean ARAP error, indicating effective shape preservation with stable and smooth vertex transitions.

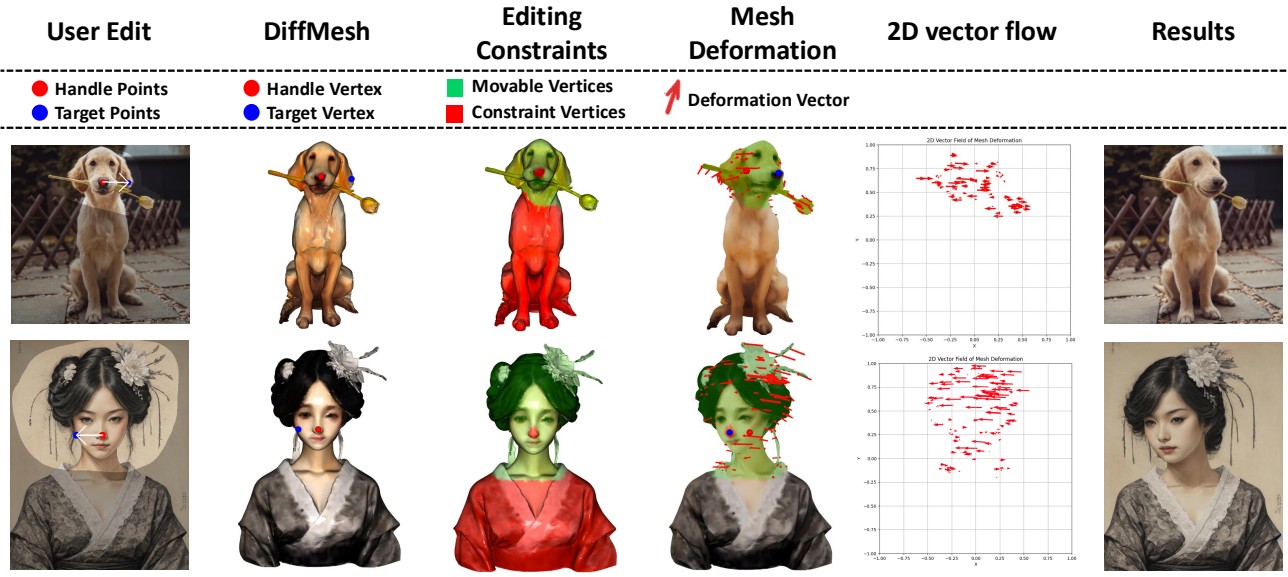

*Figure 9.* **Visualization of the mesh-guided editing process using DiffMesh.** Our FlowDrag process begins with user-defined edits specifying handle and target points. DiffMesh reconstructs a structured 3D mesh, from which editing constraints (movable and constraint vertices) are defined. Guided by these constraints, the mesh undergoes deformation, yielding clear deformation vectors. These deformation vectors are projected onto a 2D vector flow field, where representative vectors are sampled. These sampled vectors provide accurate geometric guidance, resulting in stable and precise image editing outcomes.

*Table 3.* Comparison of sampling strategies using MD and 1-LPIPS metrics on VFD-Bench.

| Sampling Strategy | MD $\downarrow$ | 1-LPIPS $\uparrow$ |
|---|---|---|
| Uniform Sub-sampling | 24.75 | 0.83 |
| Magnitude-based Sampling | **24.51** | **0.85** |

**Effect of Selected Vector Count on Performance**   We analyzed the impact of the number of selected vectors in $\hat{\Phi}$ on the PSNR metric. Fig. 8 shows the results, indicating the best performance at 10 vectors. These results emphasize the importance of selecting an optimal vector count for effective editing performance.

**Magnitude-based vs. Uniform Sub-sampling in 2D Vector Flow Field Sampling**   We compare magnitude-based sampling and uniform sub-sampling using Mean Distance (MD) and Image Fidelity (1-LPIPS). As shown in Table 3, magnitude-based sampling consistently achieves lower MD and higher 1-LPIPS scores, indicating more effective vector selection. This highlights the benefit of prioritizing vectors with larger magnitudes in the flow field.

**More Visualization Results**   Detailed visualizations of FlowDrag's mesh-guided editing process are shown in Fig.9 and Fig.16. Additional sensitivity analysis on how variations in mesh deformation influence editing outcomes can be found in Appendix D, and further analysis of mesh deformation efficiency is provided in Appendix E.

## 7. Limitation and Future Work

FlowDrag effectively addresses geometric inconsistencies prevalent in current drag-based editing methods, but several limitations remain. First, our method relies on stable mesh deformation, inherently limiting feasible dragging distances. Thus, our method is optimal for moderate dragging operations that preserve structural coherence. Additionally, FlowDrag primarily supports rigid edits, but struggles with significant content creation or removal tasks requiring major structural changes. Lastly, FlowDrag projects 3D mesh deformation onto a 2D plane, inherently losing detailed 3D structural information. This issue is compounded by FlowDrag's reliance on Stable Diffusion, which lacks 3D understanding. While our approach enhances geometric coherence, it cannot fully preserve the original 3D geometry. We believe future work could benefit from exploring inherently 3D-aware or motion-aware diffusion models, such as video diffusion, to better capture object dynamics and 3D structures, enhancing 3D understanding in image editing.

## 8. Conclusion

In this paper, we propose **FlowDrag**, a framework designed to address geometric inconsistencies in drag-based image editing via 3D mesh deformation. Additionally, we introduce **VFD-Bench**, a benchmark providing explicit ground-truth edits. Our experiments demonstrate FlowDrag's superior editing quality and geometric coherence on both Drag-Bench and VFD-Bench.

## Impact Statement

Drag-based image editing offers an intuitive and precise way to alter images, greatly enriching creative workflows. However, this technology also raises concerns about authenticity and potential misuse in generating deceptive or manipulated media. Ensuring transparency through watermarking and responsible deployment is crucial to mitigate risks while harnessing its full creative potential.

## Acknowledgements

This work was supported by Institute for Information & communications Technology Planning & Evaluation (IITP) grant funded by the Korea government(MSIT) (No.RS-2021-II211381, Development of Causal AI through Video Understanding and Reinforcement Learning, and Its Applications to Real Environments) and partly supported by Institute of Information & communications Technology Planning & Evaluation (IITP) grant funded by the Korea government(MSIT) (No.RS-2022-II220184, Development and Study of AI Technologies to Inexpensively Conform to Evolving Policy on Ethics).

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

# A. A comparison of progressive SR-ARAP performance in mesh deformation

By varying the parameter $\beta$ of the Inter-Step Smoothness term in our progressive SR-ARAP formulation (Eq.9), we evaluate how effectively the deformation preserves original geometry. To quantitatively assess these characteristics, we define two rigidity metrics: the Mean Edge Length Ratio (MELR) and the Mean ARAP Error ($\mathrm{mARAP_{Error}}$) (see SectionA.1 for detailed definitions). The MELR measures the average ratio of edge lengths in the deformed mesh relative to their original lengths, indicating how well the deformation preserves the original geometry. The ARAP Error measures a simplified residual per triangle, reflecting how closely local rotations match between the original and deformed meshes. We measure these metrics for $\beta$ values ranging from 0.2 to 1.0 to observe their effects on deformation quality. As shown in Table 4, $\beta = 0.8$ effectively balances smooth transitions and rigidity preservation, yielding stable deformations with minimal distortions.

## A.1. Rigidity Metrics

To quantify how closely a deformed mesh $\hat{M} = (\hat{V}, F)$ preserves the original geometry $M = (V, F)$, we compute two metrics specifically for the vertices optimized through the energy function (i.e., movable vertices), as constraint vertices remain fixed during deformation.

**Mean Edge Length Ratio (MELR).** Let $\{(i, j)\}$ be the set of unique edges connecting movable vertices derived from the triangular faces in $M$. For each such edge $(i, j)$, we measure the ratio of its length in the deformed mesh to its length in the original mesh, as defined in Eq. 11.

$$\mathrm{Ratio}_{ij} = \frac{\|\hat{v}_j - \hat{v}_i\|}{\|v_j - v_i\|},$$

$$\text{where } v_i, v_j \in V_{\text{movable}}, \ \hat{v}_i, \hat{v}_j \in \hat{V}_{\text{movable}}, \tag{11}$$

We then compute the mean edge length ratio (MELR), across all movable edges, as shown in Eq. 12. A MELR value closer to 1.0 indicates minimal distortion of edge lengths.

$$\mathrm{MELR} = \frac{1}{|\mathcal{E}_{\text{movable}}|} \sum_{(i,j) \in \mathcal{E}_{\text{movable}}} \mathrm{Ratio}_{ij}, \tag{12}$$

**ARAP Error.** To measure local distortion between original and deformed meshes, we compute a best-fit rotation matrix $\mathbf{R}$ for each face $\triangle = (i_0, i_1, i_2) \in F$. Let the vertex coordinates of the original and deformed face be $p_k$ and $\hat{p}_k$,

*Table 4.* Ablation study on the regularization parameter $\beta$ in Progressive SR-ARAP. A higher MELR and lower $\mathrm{mARAP_{Error}}$ indicate better preservation of mesh rigidity.

| $\beta$ | MELR($\uparrow$) | $\mathrm{mARAP}_{Error}(\downarrow)$ |
|---|---|---|
| 0.2 | 0.87 | 17.24 |
| 0.4 | 0.88 | 14.91 |
| 0.6 | 0.92 | 12.56 |
| 0.8 | **0.94** | **10.12** |
| 1.0 | 0.93 | 11.60 |

respectively, for $k \in \{0, 1, 2\}$ (Eq. 13). We first compute their centroids, $c$ and $\hat{c}$, as follows:

$$p_k, \ \hat{p}_k \quad \text{for} \quad k \in \{0, 1, 2\}, \tag{13}$$

$$c = \frac{p_0 + p_1 + p_2}{3}, \quad \hat{c} = \frac{\hat{p}_0 + \hat{p}_1 + \hat{p}_2}{3}, \tag{14}$$

Then, we form local point sets centered by these centroids:

$$P = \begin{bmatrix} p_0 - c \\ p_1 - c \\ p_2 - c \end{bmatrix}, \quad \hat{P} = \begin{bmatrix} \hat{p}_0 - \hat{c} \\ \hat{p}_1 - \hat{c} \\ \hat{p}_2 - \hat{c} \end{bmatrix}, \tag{15}$$

Next, the optimal rotation $\mathbf{R}$ aligning $P$ to $\hat{P}$ is obtained via the SVD-based Kabsch algorithm. Using this rotation, the ARAP residual for each face $\triangle$ is calculated as:

$$\mathrm{Err}_\triangle = \sum_{k=0}^{2} \|\mathbf{R}(p_k - c) - (\hat{p}_k - \hat{c})\|^2. \tag{16}$$

We then compute the overall ARAP Error by summing residuals across all faces and normalize it by the total number of faces $|F|$ to obtain the Mean ARAP Error:

$$\mathrm{mARAP_{Error}} = \frac{1}{|F|} \sum_{\triangle \in F} \mathrm{Err}_\triangle. \tag{17}$$

A lower $\mathrm{mARAP_{Error}}$ indicates better preservation of local rigidity in the deformation, enabling fair comparisons regardless of mesh size or face count.

# B. 3D Mesh Generation Based on Depth Approach

The depth-map-based 3D mesh generation discussed in Section 4.1 is formalized as the algorithm shown in Algorithm 1. We set the depth threshold ($\tau_d$) to 0.1 and defined the background threshold ($\tau_b$) as the mean depth value plus 0.3. This approach effectively removes background noise.

**Algorithm 1** Depth Map and Mesh Generation

---

**input** $\mathcal{D}$: Depth map (normalized to $[0, 1]$), $\tau_d$: Depth threshold, $\tau_b$: Background threshold

**output** $\mathcal{M} = (\mathcal{V}, \mathcal{F})$, $\mathcal{M}$: Generated 3D mesh, $\mathcal{V}$: Set of vertex, $\mathcal{F}$: Set of facets

  **Step 1: Depth Map Extraction**
    $\mathcal{D} \leftarrow \text{Marigold}(\mathcal{I})$: Extract depth map using Marigold
  **Step 2: Vertex Mapping**
    **For** each pixel $(x, y) \in \mathcal{D}$ **do**
      $z \leftarrow \mathcal{D}(x, y)$: Map depth value to $z$-coordinate
      Add vertex $V(x, y, z)$ to $\mathcal{M}$
  **Step 3: Facet Formation**
    **For** each vertex $V(x, y, z) \in \mathcal{M}$ **do**
      if $|\mathcal{D}(x, y) - \mathcal{D}(\text{adjacent})| < \tau_d$
        Connect $V$ to adjacent vertices, add facet to $\mathcal{F}$
  **Step 4: Artifact Reduction**
    **For** each facet $(v_i, v_j, v_k) \in \mathcal{F}$ **do**
      if $|\mathcal{D}(v_i) - \mathcal{D}(v_j)| \geq \tau_d$ or $\mathcal{D}(v_i) < \tau_b$
        Remove $(v_i, v_j, v_k) \in \mathcal{M}$
**Return:** $\mathcal{M} = (\mathcal{V}, \mathcal{F})$: Return the final 3D mesh

---

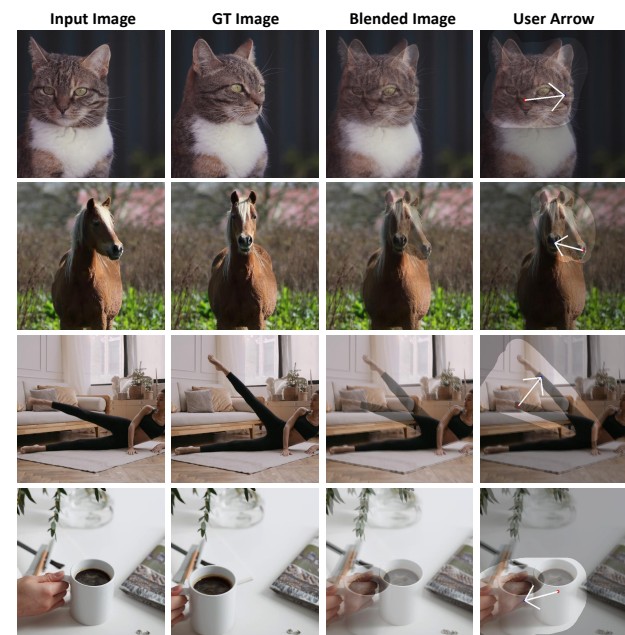

*Figure 10.* **Visualization of the labeling procedure in the VFD Dataset.** We first blend the input and ground-truth (GT) images to identify structural differences. Then, user-defined arrows are drawn, indicating precise handle points (arrow tails) and corresponding target points (arrowheads).

## C. VFD Dataset

### C.1. Dataset Categories

We constructed the VFD Dataset by selecting pairs of video frames from various sources, including DAVIS (Pont-Tuset et al., 2017), LOVEU-TGVE (Wu et al., 2023a), TVR (Lei et al., 2020), and copyright-free clips available on Pexels. Each pair captures significant structural or pose changes suitable for drag-based editing. The dataset comprises a total of 250 image pairs categorized as follows:

**Animal (140 images).** Captures diverse animal movements and expressions, including general motion changes, head movements, and mouth opening or closing.

**Human (65 images).** Includes human-related changes categorized into facial expressions, head rotations, and overall pose adjustments involving face, hand, and arm movements.

**Object (45 images).** Consists of various object movements, including household items (e.g., coffee cups, kettles), food-related actions (e.g., pizza movement, tomato slicing), and miscellaneous objects (e.g., trains, clocks).

### C.2. Labeling Procedure

To accurately define user arrows between input and ground-truth (GT) images extracted from videos, we first blend the two images and then draw user-defined arrows indicating the desired handle and target points (Fig. 10). Using this labeling method, we produce precise annotations, facilitating reliable evaluation of drag-based editing performance.

## D. Sensitivity Analysis

We perform sensitivity analysis to evaluate the robustness of mesh deformation and its subsequent effects on drag-based editing results. Specifically, we examine how variations in mesh construction parameters influence the editing quality for both DepthMesh and DiffMesh methods (Fig. 15).

First, we analyze DepthMesh by varying its reduction ratio, which determines the density of facet connections during mesh construction (Supplementary Algorithm 1, Step 3). A reduction ratio of 1 corresponds to a fully detailed mesh, while lower values progressively simplify geometry. As illustrated in Fig.15(a), we quantify the editing robustness using the PSNR ratio and 1–LPIPS ratio, comparing results to the baseline (reduction ratio = 1). Our results show that the editing performance remains robust and stable within the reduction ratio range from 1 down to 0.001, indicating that our method effectively preserves essential geometric information even under significant mesh simplification. Qualitative results (Fig.15(b)) visually confirm that drag-based editing remains accurate and stable across these reduction ratios, while excessively simplified meshes (reduction ratio = 0.0001) yield distorted and unsatisfactory results due to insufficient geometric information.

Next, we examine DiffMesh sensitivity by adjusting the sampling steps in the image-to-3D diffusion process (Hun-

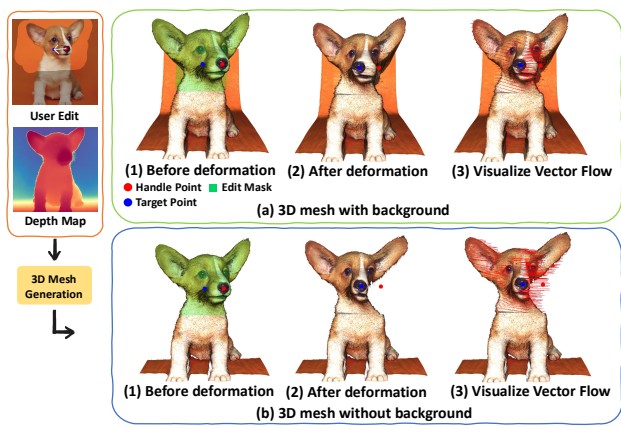

Figure 11. **DepthMesh deformation results when generating the mesh from the depth map with and without background consideration.** (a) Result with background. (b) Result without background.

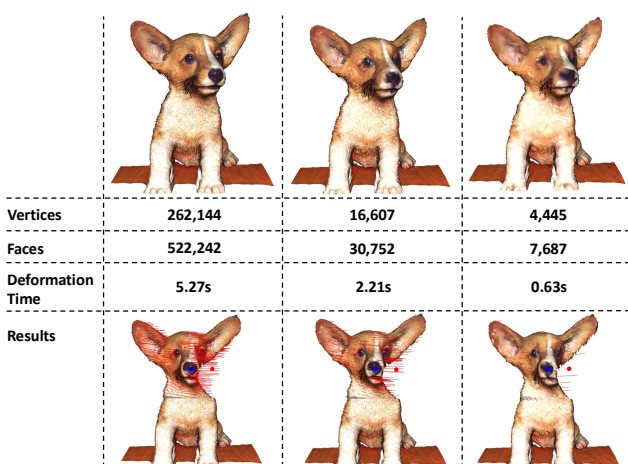

Figure 12. **Deformation time and results based on the number of vertices and faces in the 3D Mesh.** As vertices and faces decrease, deformation time also decreases, but the ability to capture detailed mesh changes is reduced.

yuan3D 2.0 (Zhao et al., 2025)). As demonstrated in Fig.15(c), editing quality is robust across sampling steps ranging from 10 to 40, ensuring consistent geometric guidance. Qualitative evaluation (Fig.15(d)) further verifies that our method maintains stable and accurate editing performance for sampling steps of 10 or higher. However, fewer sampling steps (e.g., step = 5) degrade the mesh geometry, resulting in noticeable quality deterioration in drag editing outcomes.

These analyses collectively confirm that our method demonstrates robustness and stability in drag-based editing across a wide range of mesh deformation parameters.

## E. Analysis of Mesh Deformation Efficiency

### E.1. 1. Impact of Background Separation on Mesh Deformation Quality

Unlike DiffMesh, which inherently isolates foreground objects due to the diffusion generation process, the quality of DepthMesh deformation can significantly vary depending on whether the background is included. To analyze this effect, we performed mesh deformation experiments both with and without background separation. The results are presented in Fig. 11. As shown in Fig. 11(a), when the background is included in the mesh, the deformation appears constrained and unnatural due to the influence of the background. In contrast, Fig. 11(b) shows that when the background is removed, the mesh deforms much more freely and naturally. Based on these observations, we decided to focus on foreground meshes only, excluding the background, when generating the mesh for deformation.

### E.2. 2. Deformation Time Relative to Face Count

Mesh deformation is a critical step in FlowDrag, as it generates the 2D vector flow field required for input. To assess the time required for mesh deformation, we measured the deformation time while adjusting the number of faces in the mesh. In Fig. 12, the sample on the far left represents a mesh generated directly from the depth map of the image, with a deformation time of 5.27 seconds. As we move to the right, the number of faces in the mesh decreases, and we observe a corresponding reduction in deformation time. However, reducing the number of faces also diminishes the mesh's ability to capture fine deformations accurately. Therefore, to better preserve the fidelity of the mesh transformations, we opted not to reduce the face count during deformation. On average, this approach resulted in a processing time of 5 seconds per sample.

## F. Additional Qualitative Comparisons

Additional qualitative results on DragBench and VFD-Bench are presented in Fig.13 and Fig.14, respectively. FlowDrag consistently preserves spatial coherence and produces more stable editing outcomes compared to existing drag-based methods, demonstrating the benefits of leveraging structured 3D spatial context.

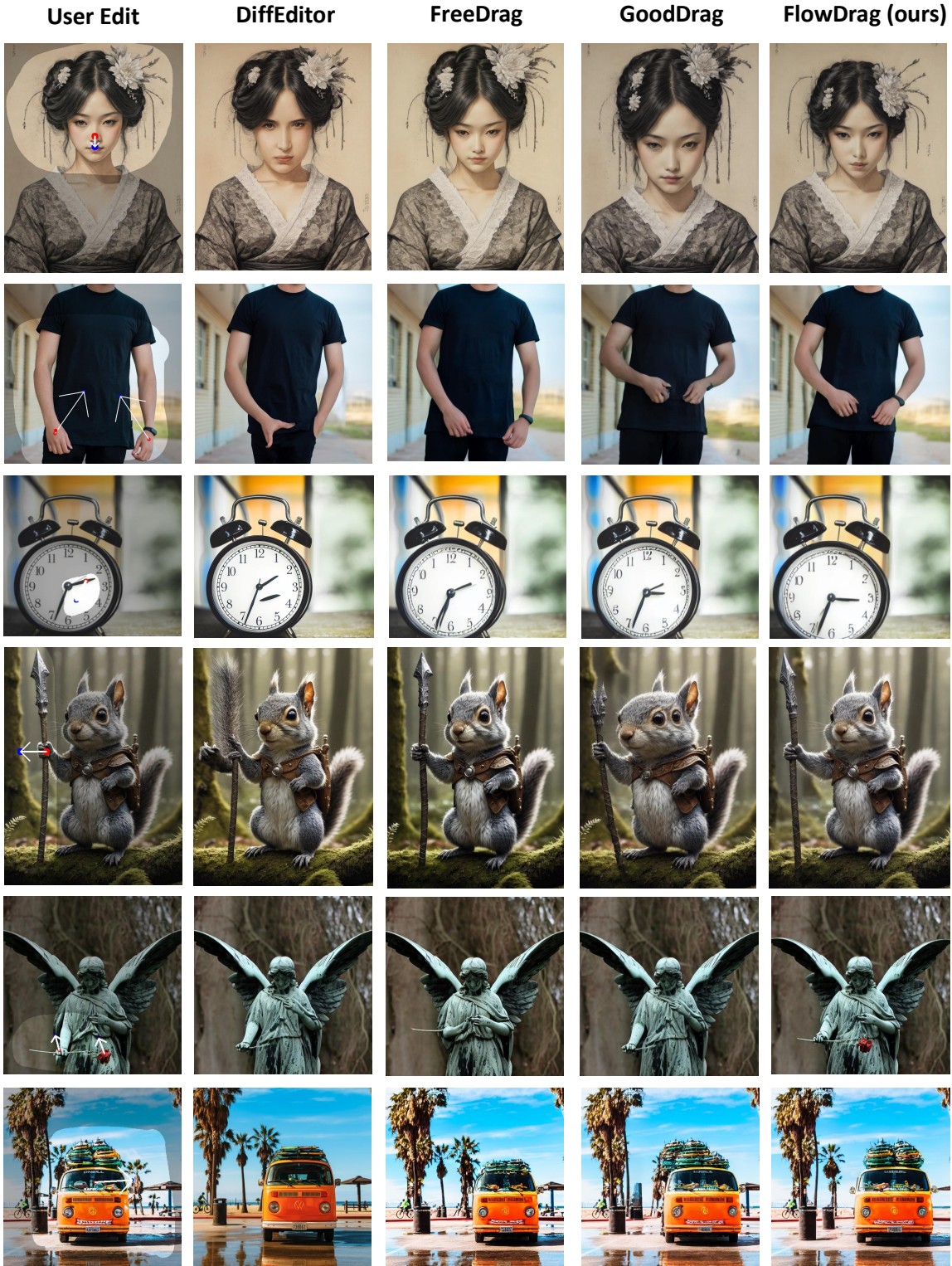

*Figure 13.* **Additional Qualitative Comparisons on DragBench.**

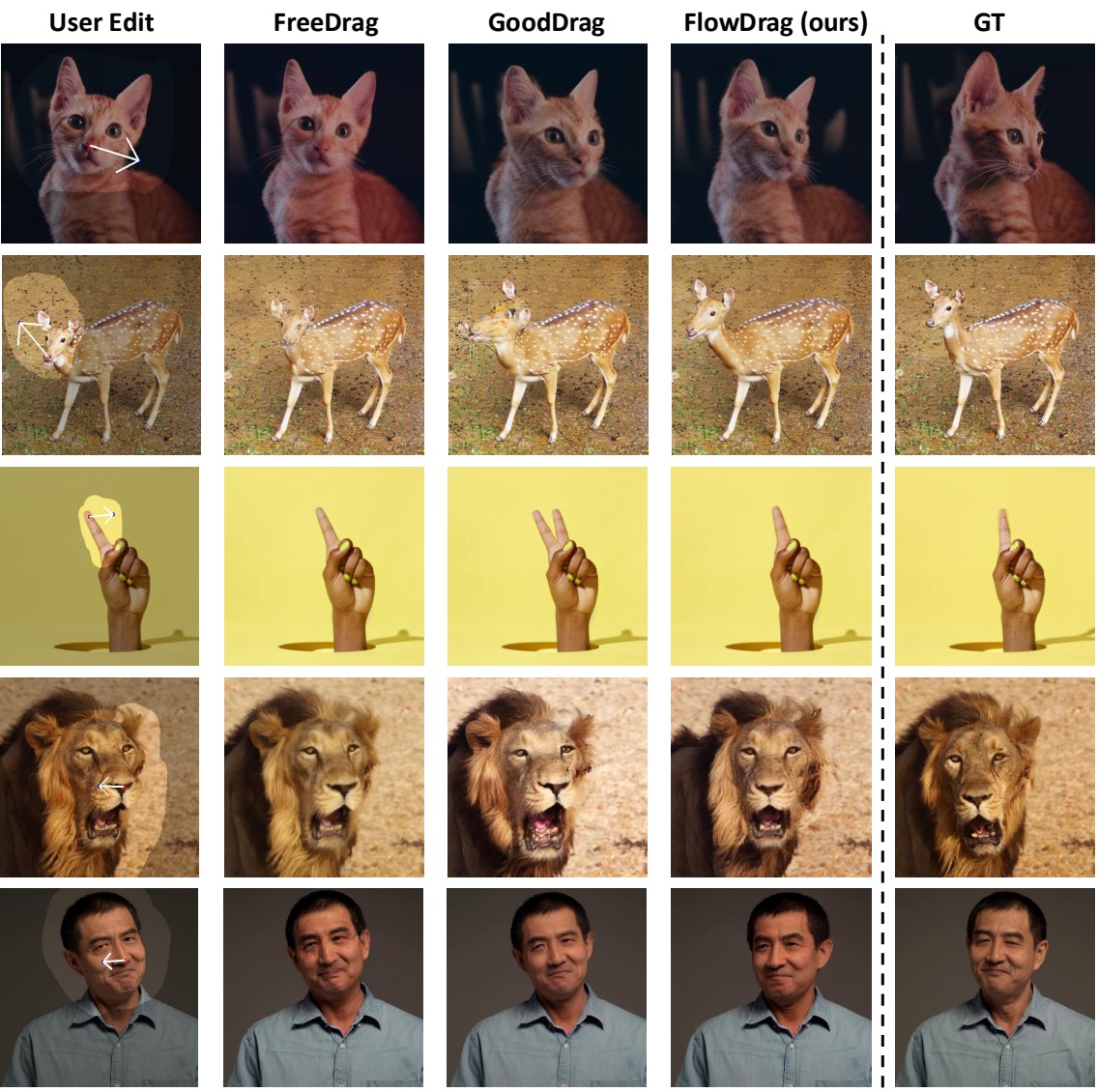

*Figure 14.* **Additional Qualitative Comparisons on VFD-Bench.**

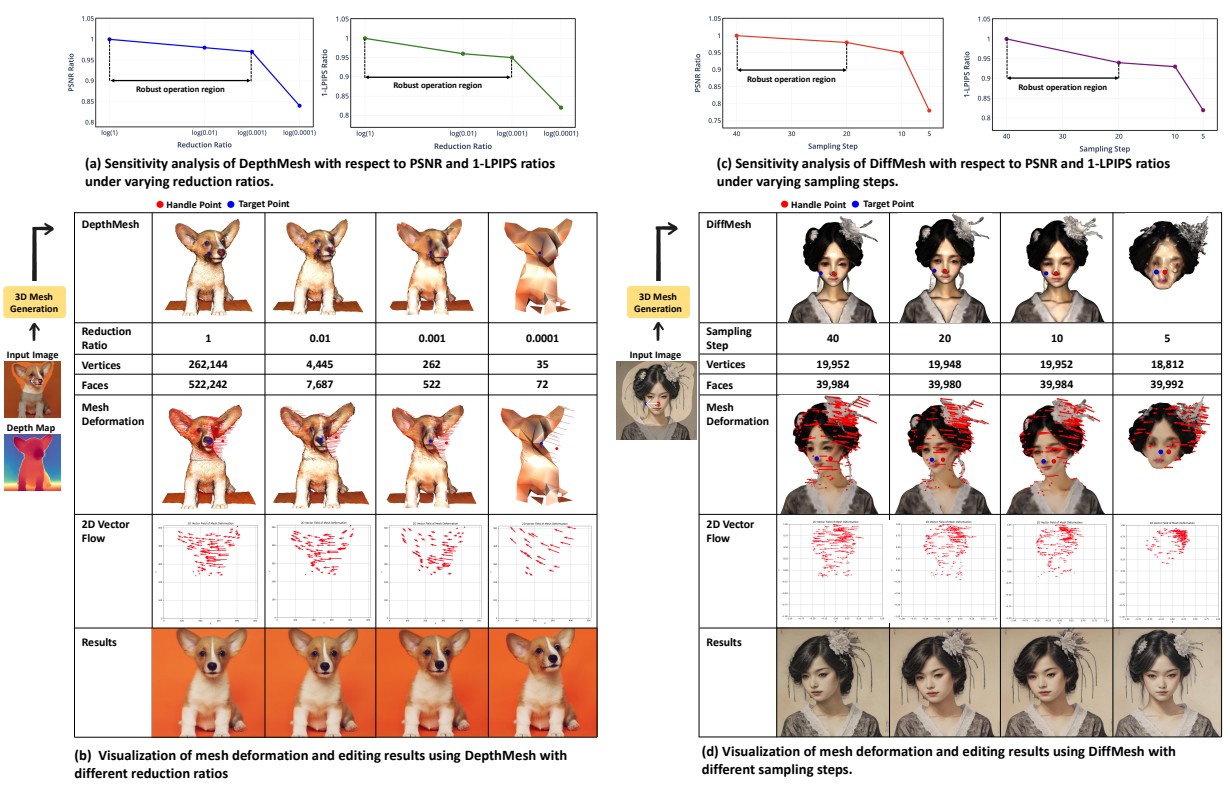

*Figure 15.* **Sensitivity analysis and qualitative comparison of mesh deformation and drag-editing results.** (a) We evaluate the robustness of DepthMesh under different reduction ratios, which control the density of facet connections during mesh construction (see Algorithm 1). A ratio of 1 denotes a fully connected mesh, while lower values lead to degraded geometry. Since no ground-truth exists for DragBench dataset, we compute the PSNR ratio and 1–LPIPS ratio using the result at reduction ratio = 1 as reference. The editing remains robust within the reduction ratio range of 0.001 to 1. (b) Qualitative results confirm that our method maintains stable and accurate editing across the reduction ratio range of 0.001 to 1. (c) We assess the robustness of DiffMesh by varying the sampling step in the image-to-3D diffusion model (Hunyuan3D 2.0 (Zhao et al., 2025)). Editing quality remains stable for sampling steps of 10 to 40. (d) Visual results show that our method remains robust for sampling steps of 10 or higher.

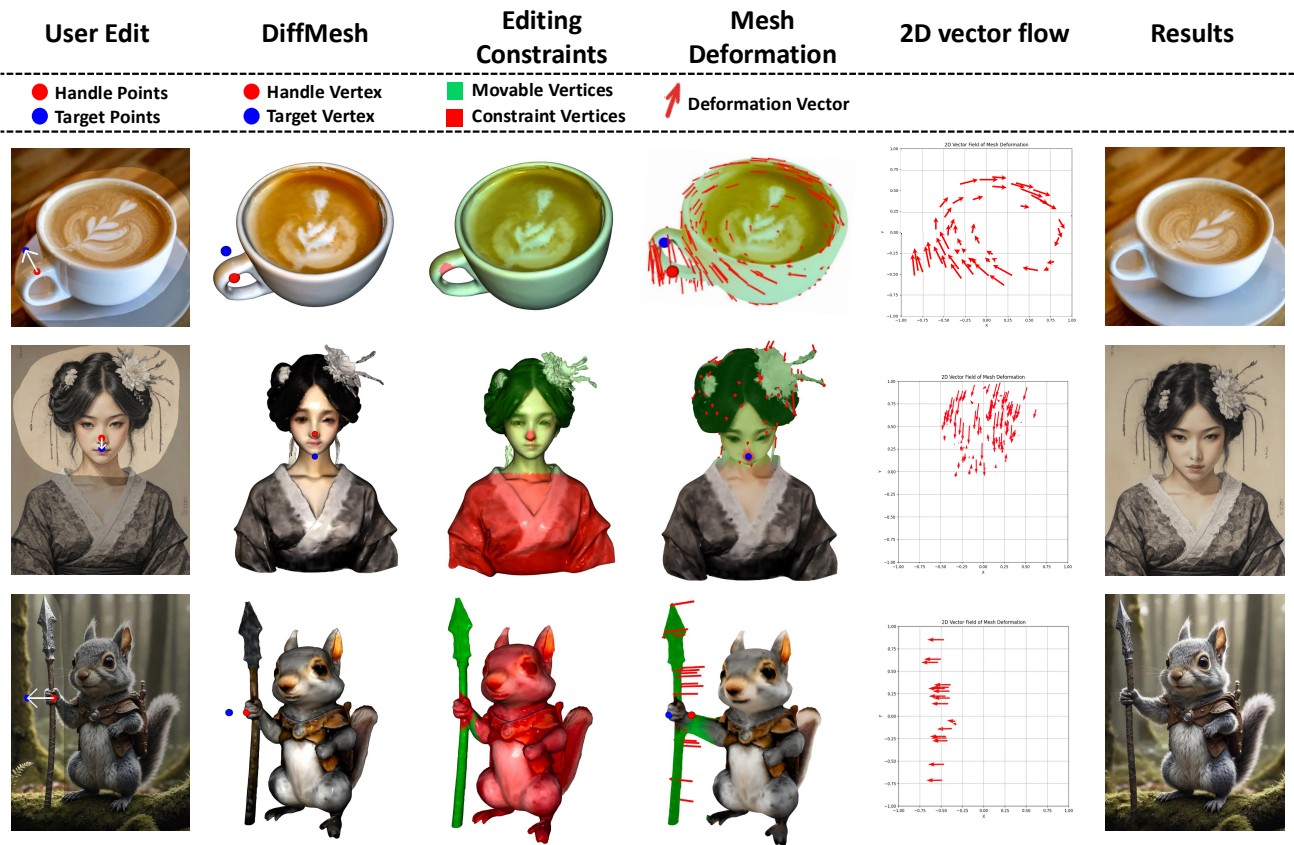

*Figure 16.* **More Visualization of the mesh-guided editing process using DiffMesh.**

