# OpenReview forum: "FlowDrag: 3D-aware Drag-based Image Editing with Mesh-guided Deformation Vector Flow Fields"
_ICML.cc/2025/Conference — ICML 2025 spotlightposter_

### Official Review · Reviewer_bFUV · 2025-03-13

**Overall Recommendation:** 4

**Summary:**

This paper proposes a novel method for drag-based image editing. Compared with the previous work, the proposed method takes the 3D geometric information into consideration through mesh construction, ensuring a stable and 3D-plausible editing. The method is claimed to achieve state-of-the-art performance.

**Claims And Evidence:**

- The method is not comparing with the very first work of this task: DragGAN. Even though the proposed method is diffusion-guided, DragGAN is still observed achieving better performance in certain cases. This hurts the soundness of this paper.
    - As a comparison: GoodDrag compares with DragGAN.
- The paper shows some high-quality qualitative results. This can demonstrate the claim of state-of-the-art performance.
- However, this paper is not using Drag100 benchmark, which is created in the most powerful baseline GoodDrag, and contains various types of editing as shown in GoodDrag's Fig.6 (editing involves content removal and creation). This raises lots of concerns about whether the proposed method can also achieve this.

**Essential References Not Discussed:**

-  As mentioned in "Claims And Evidence", the method is not comparing with the very first work of this task: DragGAN, just cited once.

**Experimental Designs Or Analyses:**

-  As mentioned in "Claims And Evidence"
    - The method is not comparing with the very first work of this task: DragGAN. Even though the proposed method is diffusion-guided, DragGAN is still observed achieving better performance in certain cases. This hurts the soundness of this paper.
    - Drag100 benchmark proposed in the paper of the most powerful baseline GoodDrag is not used for experiments. This raises concerns about whether the proposed method can support editing involving content removal and creation.

**Methods And Evaluation Criteria:**

- The method is reasonable and well-motivated. I am especially impressed by the progressive deformation part.
- As mentioned in "Claims And Evidence", Drag100 benchmark proposed in the paper of the most powerful baseline GoodDrag is not used for experiments. This raises concerns about whether the proposed method can support editing involving content removal and creation.
    - In fact, from the method, I think the proposed method cannot well support content creation as it will be hard to generate the new contents in 3D. I would like the author to show some results to disprove this.

**Other Comments Or Suggestions:**

I would like to see some additional results (1) comparing with DragGAN, and (2) from the "content removal" or "content creation" categories of Drag100 dataset.

All these results are received in rebuttal, resolving most of my concerns. Therefore, I will raise my score from 3 to 4.

**Other Strengths And Weaknesses:**

- The two orange arrows in Fig.4 are overlapped with texts, which makes the figure quite messy.
- There is no link in citations and references to figures and tables.
- The axis metrics in Fig.2 (d) and (e) are inconsistent.

**Questions For Authors:**

Please refer to the reviews above.

**Relation To Broader Scientific Literature:**

This paper proposes a novel approach to inject 3D awareness and basis to drag-based editing pipelines. This enables the requirement of 3D-plausibility in these editing tasks and augments the capability.

**Theoretical Claims:**

N/A.

---

> ### Author Rebuttal · Authors · 2025-04-01
>
> **[Q1] The paper lacks comparison with DragGAN, which achieves better performance in certain cases.**
>
> **[A1]** We provide additional qualitative comparisons between FlowDrag and DragGAN in Fig. 14(a) (please refer to the link below). To ensure fair comparisons, we reproduced DragGAN using the official GitHub repository and employed PTI [1] for GAN inversion. Since the provided DragGAN weights are limited to the StyleGAN-Human model, our comparisons focus exclusively on human images. As shown in Fig. 14(a), DragGAN struggles to preserve the original person's identity, whereas FlowDrag maintains identity effectively and exhibits better geometric consistency.
>
> Fig 14. https://anonymous.4open.science/r/FlowDrag-B950/Fig_14.pdf
>
> [1] Pivotal Tuning for Latent-based Editing of Real Images (ACM TOG 2022)
>
> **[Q2] I would like to see additional results from the "content removal" or "content creation" categories of the Drag100 dataset.**
>
> **[A2]** We applied FlowDrag to the "content creation" and "content removal" examples from the Drag100 dataset, and present additional results in Fig. 14(b)-(c). In the "content creation" category (Fig. 14(b)), FlowDrag successfully generates natural results with fewer artifacts compared to GoodDrag. Similarly, in the "content removal" category (Fig. 14(c)), FlowDrag demonstrates editing quality that is comparable or superior to GoodDrag.
> Although FlowDrag primarily targets "rigid edits" to preserve object rigidity during editing (as stated in our introduction), these results confirm that our method is also effective in handling non-rigid editing scenarios.
>
> **[Q3] The two orange arrows in Fig.4 overlap with text, making the figure quite messy.**
>
> **[A3]** Thank you for pointing this out. We will adjust the positioning of the orange arrows in Fig.4 to avoid overlap and improve visual clarity in the final manuscript.
>
>
> **[Q4] There is no link in citations and references to figures and tables.**
>
> **[A4]** We will include hyperlinks in citations and references to figures and tables in the final manuscript to enhance readability.
>
>
> **[Q5] The axis metrics in Fig.2 (d) and (e) are inconsistent.**
>
> **[A5]** Yes, the axis metrics in Fig. 2(d) and Fig. 2(e) intentionally differ. Fig. 2(d) compares methods on the DragBench dataset using metrics such as 1-LPIPS (for image fidelity) and MD (Mean Distance, for evaluating handle-point movement accuracy). In contrast, Fig. 2(e) shows comparisons on our proposed VFD-Bench dataset, which provides ground-truth edited images from video frames, enabling evaluation using RGB-level (PSNR) and feature-space metrics (1-LPIPS and MD). Fig. 2(e) specifically plots results using PSNR and 1-LPIPS. We will explicitly clarify this distinction in the final manuscript.

---

### Official Review · Reviewer_GpMK · 2025-03-13

**Overall Recommendation:** 4

**Summary:**

This paper proposes FlowDrag, a method that leverages pre-trained stable diffusion models for drag-based image editing. This method improves the drag-based image editing by building a field of 3D-aware dragging instruction from the user's input. Specifically, FlowDrag first leverages an image-to-depth or image-to-mesh model to generate a mesh for the foreground object. It then applies SR-ARAP, a mesh deformation algorithm, to calculate how the user's drag deformed the object. Through the novel progressive deformation with SR-ARAP, it obtains a flow field showing how a larger area of the object would move. This flow field is then sampled and projected to 2D space to act like the densified dragging instruction. By providing a field of 3D-aware dragging instructions, the diffusion model receives more guidance on how each pixel is going to move and yields a better editing result with the standard motion supervision and point tracking pipeline. Moreover, FlowDrag further uses the projected deformed mesh as a guide to improve the editing result. It also constructs a new drag-based image editing benchmark, VFD-Bench Dataset, with ground-truth editing results to compensate for the fact that the existing benchmark, DragBench, does not provide ground-truth images. FlowDrag is evaluated on both DragBench and VFD-Bench datasets and shows superior performance than previous methods.

## Update after rebuttal
I appreciate the author's response and insightful discussion. I would keep my original rating.

**Claims And Evidence:**

There are two main contributions claimed by the paper: the densification of dragging instruction through mesh deformation and VFD-Bench datasets. Both of them are well substantiated by the experiment results. The effectiveness of FlowDrag is demonstrated by the results in Table 1, Table 2, and Table 3. Each component in the method is studied in the ablation study. The VFD-Bench dataset provides ground-truth images for drag-based image editing, which is exactly what the community needs, as it provides an accurate and objective way to evaluate the result of image editing. Therefore, the claims in this paper are very solid.

**Essential References Not Discussed:**

All related works have been discussed.

**Experimental Designs Or Analyses:**

The experiments are thorough and detailed, with no significant issue. Given that the flow field generation is independent of a specific editing method, a potential improvement in the experiment could be applying the flow field to a range of drag-based editing methods, such as DragDiffusion, GoodDrag, etc, to see whether the flow field may improve existing methods. If it could, the significance of this work would be much greater.

**Methods And Evaluation Criteria:**

The proposed method is a novel and clear way to improve the drag-based image editing using diffusion models. It realized that a good editing result should comply with local 3D rigidity constraints. However, the user's input is sparse and existing methods rely on the inference ability of the diffusion models to hallucinate how the object may change. This could be inaccurate as the diffusion models are trained on pure 2D data so it has no idea how the object should deform in 3D. This paper chooses an intuitive and explicit way to achieve 3D-aware editing: lifting the object from 2D to 3D, simulating the change in 3D caused by the drag, and reflecting these changes in 2D in the form of a flow field. This design targets the main problem very well, and the design is intuitive and logical.

The paper also proposes a new dataset VFD-Benchmark, a drag-based image editing benchmark with ground-truth editing result. The existing benchmark, DragBench, only provides images and instructions but no ground-truth editing results. It relies on Mean Distance (MD) and 1-LPIPS to evaluate the editing results. However, these two metrics cannot accurately reflect the editing effect: MD only compares DIFT feature similarity between source and target keypoints, and 1-LPIPS ignores object's deformations before and after editing. VFD-Benchmark provides ground-truth editing images. This allows the editing to be accurately and objectively evaluated by comparing it with the ground-truth image. The dataset offers a good solution to an existing issue in the evaluation.

Therefore, the paper provides solid solutions to both the editing pipeline and evaluation, making significant contributions to this field.

**Other Comments Or Suggestions:**

The proposed method is solid and intuitive, and the paper is well-written and easy to follow. It makes concrete contributions to the field. Therefore, I recommend the acceptance of this paper.

**Other Strengths And Weaknesses:**

Some place requires further clarifications, which I would elaborate in the Question section. These questions do not affect the contribution and significance of this work.

**Questions For Authors:**

I would appreciate it if the author could address the following questions:

1. Regarding the Progressive Deformation with SR-ARAP, what is the value of $\lambda$ in Equation 8? I am also confused with the handle-matching term in Equation 9. $v_t$ is fixed because it is the target point, $v^{(k+1)}_{h}$ is calculated based on Equation 8. Therefore, both terms seem fixed. What is the learnable in here? Is it $\lambda$?

2. The paper mentioned that both DepthMesh and DiffMesh are used, yet there is no experiment or discussion on which one is preferred. Could you provide more information on this matter? Which one is used to achieve the reported result in Table 1, 2 and 3?

3. Section C of the supplementary mentions background separation. How is it achieved?

4. What are the hardware requirement for FlowDrag?

**Relation To Broader Scientific Literature:**

The paper is built on the existing drag-based image editing pipeline. It follows motion supervision, point tracking and latent representation optimization proposed by DragDiffusion. The evaluation metrics, Mean Distance and 1-LPIPS, are commonly used by the literature.

**Theoretical Claims:**

This paper is an application paper with no significant theoretical contribution.

---

> ### Author Rebuttal · Authors · 2025-04-01
>
> **[Q1] What is the value of $\lambda$ in Equation 8?**
>
> **[A1]** The parameter $\lambda$ in Eq. (8) represents the incremental step size at each iteration, indicating the fraction of the displacement between the handle vertex ($v_h$) and the target vertex ($v_t$). However, we discovered a minor typo in the original version of Eq. (8), and we sincerely appreciate the reviewer’s careful observation. The corrected Eq. (8) is:
>
> $ v_h^{(k+1)} = v_h^{(k)} + \lambda (v_t - v_h), \quad 0 < \lambda \le 1 $
>
> In this equation, both $v_h^{(k+1)}$ and $v_h^{(k)}$ are intermediate vertices positioned incrementally between the initial handle vertex ($v_h$) and the target vertex ($v_t$).
> To facilitate a clearer understanding, we provide an enhanced illustration in Fig. 15 (please refer to the link provided below), complementing Fig. 3 in the paper.
> This formulation implies that instead of moving the handle vertex directly from $v_h$ to $v_t$ in one step, our method progressively moves it through multiple intermediate positions. Specifically, we set $\lambda=0.2$ in our implementation, meaning the vertex moves 20% closer to its target at each iteration. For example, with $\lambda=0.2$, the handle vertex path would be as follows:
>
> $v_h (\text{handle vertex}) = v_h^{0} → v_h^{1} → v_h^{2} → v_h^{3} → v_h^{4} → v_h^{5} = v_t (\text{target vertex})$
>
> This progressive SR-ARAP algorithm thus prevents abrupt mesh distortions by smoothly distributing large vertex displacements across multiple intermediate steps, achieving more stable and coherent deformations.
> We will clearly correct Eq. (8) accordingly in the revised manuscript. We sincerely thank the reviewer for this valuable clarification.
>
>
> Fig 15. https://anonymous.4open.science/r/FlowDrag-B950/Fig_15.pdf
>
>
> **[Q2] I am also confused with the handle-matching term in Equation 9.**
>
> **[A2]** The handle-matching term in Eq. (9) acts as a soft constraint, ensuring handle vertices smoothly approach their intended intermediate (and ultimately final) positions. Since the SR-ARAP algorithm primarily optimizes vertex positions by minimizing local rigidity (minimal local distortion), handle vertices may not exactly reach intermediate targets. To resolve this, the handle-matching term gently penalizes deviations from target positions:
>
> $\beta \sum_{v_h \in \text{handles}} \Bigl\|\, v_h^{(k+1)} - v_t \Bigr\|^2$
>
> This ensures balanced optimization between local rigidity and accurate vertex positioning. We will clarify this explicitly in the revised manuscript.
>
> **[Q3] $v_t$ and $v^{(k+1)}_{h}$ are fixed. What is the learnable in Eq (8)? Is it $\lambda$?**
>
> **[A3]** Yes, $v_t$ and $v^{(k+1)}_{h}$ are both fixed. Additionally, as explained in A1-2, the parameter $\lambda$ is a manually set hyperparameter (set as 0.2) and not learnable. Eq. (8) itself contains no learnable parameters. The learnable parameters are the positions of vertices (other than handle and target vertices), which are optimized through the SR-ARAP energy function with the handle-matching term in Eq. (9).
>
> **[Q4] The paper mentions using both DepthMesh and DiffMesh. Which one is preferred and why? Additionally, which mesh (DepthMesh or DiffMesh) is used to achieve the reported results in Tables 1, 2, and 3?**
>
> **[A4]** We provide additional experimental comparisons of drag-editing results using DepthMesh and DiffMesh in Fig. 12 (please refer to the link provided below).
> As shown in Fig. 12(a), DepthMesh struggles to generate geometry for regions unseen in the single input image, resulting in unnatural mesh deformation. In contrast, DiffMesh (Fig. 12(b)), generated via a diffusion model, effectively captures complete geometry, enabling more natural and coherent mesh deformation. Therefore, we prefer DiffMesh due to its better geometric consistency, crucial for accurate mesh deformation using our SR-ARAP algorithm.
> All reported results in Tables 1, 2, and 3 are based on DiffMesh. The detailed editing process using DiffMesh is illustrated in Fig. 11.
>
> Fig 11. https://anonymous.4open.science/r/FlowDrag-B950/Fig_11.pdf
>
> Fig 12. https://anonymous.4open.science/r/FlowDrag-B950/Fig_12.pdf
>
>
> **[Q5] Section C of the supplementary mentions background separation. How is it achieved?**
>
> **[A5]** Background separation is performed during the DepthMesh generation by applying a background threshold ($τ_b$) in Step 4 of Algorithm 1 (Supplementary, Section B). Specifically, any mesh facets with depth values smaller than $τ_b$ are removed. For example, Fig. 7(a) in Section C illustrates a result without background separation ($τ_b=0$), whereas Fig. 7(b), with $τ_b=0.3$, effectively removes background facets.
>
> **[Q6] What are the hardware requirement for FlowDrag?**
>
> **[A6]** FlowDrag requires less than 14GB of GPU memory for processing a 512×512 input image, as evaluated on a single NVIDIA A100 GPU. We will include detailed hardware requirements in the final manuscript.

---

> > ### Comment · Reviewer_GpMK · 2025-04-04
> >
> > Thank you for your detailed response. So the purpose of the regularization term is to correct the position of $v^{(k+1)}_{h}$ because the moving direction $(v_t - v_h)$ in Equation 8 is solely based on initial and target positions, so this direction does not always moving point towards target positions. Therefore, the regularization term is included to move points towards target positions from current positions? If that is the case, isn't the original version of Equation 8 better because its updated direction is calculated based on the current positions?

---

> > > ### Author Response · Authors · 2025-04-05
> > >
> > > Thank you for your thoughtful comments. Your understanding of the regularization term is indeed correct. In our earlier rebuttal [A1], we presented a modified Eq. (8) with a fixed $\lambda$. However, as you suggested, the original Eq. (8) calculates the update direction based on the current position (intermediate vertex), thus representing a more general and reasonable approach. In this case, dynamically adjusting $\lambda$ via a scheduling strategy can also be effective to ensure that vertices reliably reach the target position. Following your valuable feedback, we will present both approaches (fixed vs. dynamic $\lambda$) in our final manuscript and provide additional comparative experiments. Thank you!

---

### Official Review · Reviewer_JQSi · 2025-03-14

**Overall Recommendation:** 3

**Summary:**

This paper proposes a novel drag-based editing framework called FlowDrag. Its key feature is the introduction of control points generated through the deformation of a 3D mesh, which helps to mitigate the geometric discontinuities commonly present in existing drag-based editing methods. Judging from the results provided in the paper, this method is quite effective and shows a noticeable improvement over current approaches.

**Claims And Evidence:**

The authors introduced a dedicated dataset for testing called VFD-Bench, which provides a more comprehensive quantitative analysis than existing methods. However, the paper presents a limited number of visualization results, and a qualitative analysis should include more visual effects.

**Essential References Not Discussed:**

None

**Experimental Designs Or Analyses:**

Yes

**Methods And Evaluation Criteria:**

Yes

**Other Comments Or Suggestions:**

Refer to Weaknesses

**Other Strengths And Weaknesses:**

The paper introduces a 3D Mesh model to determine the control points needed for editing, which is innovative。The quantitative and qualitative analysis results provided demonstrate the feasibility and effectiveness of the approach.
Weaknesses:
However, the authors do not discuss the impact of the 3D Mesh model on the results. It is well known that algorithms for constructing mesh models from a single image are still immature. If mesh construction fails or there are serious artifacts, could this lead to image editing failure? At the same time, regarding the selection of control points, as can be seen from Figure 3, the geometry of the edited dog's head is not consistent after mesh editing. If control points are chosen in a regular manner, could this lead to bad results?

**Questions For Authors:**

The paper should provide more visual editing results to further demonstrate the effectiveness of the proposed method. This would help to substantiate the claims made in the paper and offer a more comprehensive understanding of the technique's capabilities and potential applications.
My view is that this paper is meaningful in improving existing drag-based image editing methods from a 3D perspective, but the proof of visual effects should be strengthened. Therefore, I have recommended a "weak accept."

**Relation To Broader Scientific Literature:**

Image editing

**Theoretical Claims:**

The paper improves the SR ARAP, and provides formula derivations in the supplementary material. I have reviewed them and found no significant issues.

---

> ### Author Rebuttal · Authors · 2025-04-01
>
> **[Q1] The authors do not discuss the impact of 3D mesh construction. Could failures or severe artifacts in mesh reconstruction cause image editing to fail?**
>
> **[A1]** Yes, severe artifacts or failures in 3D mesh reconstruction could hinder the accurate generation of an accurate 2D vector flow, potentially causing editing failures. To address this concern, we evaluated the robustness of FlowDrag across various mesh reconstruction conditions, as detailed in our response to Reviewer 1 (A2) and illustrated in Fig. 13 (please refer to the link provided below). Due to the space limitations, we respectfully ask the reviewer to refer to these detailed analyses.
> Briefly, we analyzed two mesh-generation approaches used in FlowDrag: DepthMesh and DiffMesh. For DepthMesh, varying the reduction ratio (controlling mesh density during construction) can degrade the original geometry if reduced excessively. For DiffMesh, changing the diffusion sampling step of the image-to-3D diffusion model (Hunyuan3D 2.0) can introduce artifacts and geometry degradation when sampling steps become very short. Our comprehensive experiments identified robust operating ranges: DepthMesh remains robust within a reduction ratio range of approximately 0.001–1, and DiffMesh maintains robustness for diffusion sampling steps of 10 or higher.
> These analyses demonstrate FlowDrag's robustness, confirming its capability to consistently produce stable and accurate editing outcomes, even under varying conditions of mesh reconstruction quality.
>
> Fig 13. https://anonymous.4open.science/r/FlowDrag-B950/Fig_13.pdf
>
>
> **[Q2] Regarding the selection of control points, as seen in Figure 3, the geometry of the edited dog's head is inconsistent after mesh editing. If control points are chosen in a regular manner, could this lead to bad results?**
>
> **[A2]** As shown in Fig 3, minor geometric inconsistencies may arise from the mesh deformation itself, since the deformed mesh does not perfectly preserve fine-grained rigidity. However, we do not directly use this mesh for editing. Instead, we project the deformed mesh onto a 2D vector field and select optimal control points (referred to as "2D vector flow") from this projection. Furthermore, we conducted the experiment on selecting control points in a regular manner corresponding to the "Uniform sub-sampling" approach described in our paper (Sec 4.3). Our experiments show that this approach still effectively maintains overall geometric consistency and outperforms many existing methods. (We guess this effectiveness is attributed to our multiple drag vector concept.) Additionally, we proposed "Magnitude-based sampling", which selects more effective vectors with the largest displacements, achieves optimal editing results, as quantitatively demonstrated in Table 5.
>
>
> **[Q3] The paper presents limited qualitative visualization results.**
>
> **[A3]** We provide additional qualitative visualization results in Fig. 10 (additional results on VFD-Bench and Drag100), Fig. 11 (visualization of the mesh-guided editing process using DiffMesh), Fig. 12 (comparison of mesh deformation pipelines using DepthMesh and DiffMesh), Fig. 13 (sensitivity analysis and robustness comparison of mesh deformation), and Fig. 14 (additional qualitative comparisons of drag-based editing results). We will also incorporate these results in the final manuscript.
>
> Fig 10. https://anonymous.4open.science/r/FlowDrag-B950/Fig_10.pdf
>
> Fig 11. https://anonymous.4open.science/r/FlowDrag-B950/Fig_11.pdf
>
> Fig 12. https://anonymous.4open.science/r/FlowDrag-B950/Fig_12.pdf
>
> Fig 13. https://anonymous.4open.science/r/FlowDrag-B950/Fig_13.pdf
>
> Fig 14. https://anonymous.4open.science/r/FlowDrag-B950/Fig_14.pdf

---

> > ### Comment · Reviewer_JQSi · 2025-04-06
> >
> > Thank you to the authors for their detailed responses. I believe the authors' replies have basically resolved my confusion. I had already given an opinion leaning towards acceptance in the first round of review. I maintain my rating, and I am inclined to recommend the acceptance of this paper.

---

### Official Review · Reviewer_cw4s · 2025-03-14

**Overall Recommendation:** 3

**Summary:**

This paper proposes FlowDrag, which focuses on improving geometry consistency of drag-based image editing. It reconstructs a 3D mesh from the image, and uses an energy function to guide mesh deformation. The deformed mesh is then projected into 2D and used to guide the image editing denoising process. This paper also proposes a new benchmark dataset, VidFrameDrag (VFD), as the first drag-editing benchmark that has ground truths using consecutive view shots.
Experiments are conducted on both the proposed VFD benchmark and an existing DragBench and validate the effectiveness of the proposed method, evaluated by MD and 1-LPIPS as metrics and a user study.

## update after rebuttal
The rebuttal has addressed most of my concerns. I believe that maintaining my original positive rating appropriately reflects my overall favorable impression of the work.

**Claims And Evidence:**

- The motivation is reasonable and straightforward. To preserve geometric consistency, the paper adds a 3D mesh as an intermediate representation for 2D editing to inject 3D geometric prior.
- It is also great to clarify that the model specifically tackles the kind of "rigid edit," which only contains rigid transformations.

**Essential References Not Discussed:**

NA

**Experimental Designs Or Analyses:**

- The experiment results show that the proposed method outperforms existing methods, including DiffEditor, DragDiffusion, DragNoise, FreeDrag, and GoodDrag, on the VFD-Bench, evaluated by 1-LPIPS and MD.
- On the DragBench dataset, it has slightly lower 1-LPIPS values than the bests, while the paper reasonably argues it is because the other methods induce minimal edits.
- User study is also conducted to validate the method’s superior performance.

**Methods And Evaluation Criteria:**

- [Pipeline]: The pipeline is carefully designed. The input image is used to generate a 3D mesh leveraging off-the-shelf tools. The drag modification is done in the 3D space by offsetting the mesh vertices using an energy-function based method ARAP (and its followup SR ARAP). A progressive process is carefully crafted for better deformation. The deformed mesh vertices are then sampled using two candidate strategies, and used to extract a 2D vector flow map for motion supervision and point tracking, as well as used to obtain a 2D projection and inject the Unet layout features to guide the spatial and geometry information.
- [VFD-Bench]: The introduced VFD-Bench provides ground truths by leveraging video frames. This effectively provides fair evaluation capabilities and could have a positive impact on the field.

**Other Comments Or Suggestions:**

NA

**Other Strengths And Weaknesses:**

- [Fine-grained geometry inconsistency]: The method facilitates overall geometric and spatial consistency; while the edited images still showcase some inconsistency in the fine-grained geometry. For example, the hat's shape in the first sample of Fig.6.

**Questions For Authors:**

- [3D mesh quality] How important is the quality of the 3D mesh reconstruction? What will happen if the 3D mesh fails to represent the image? How often will it succeed in generation, and is it robust? More ablation and insights on this would help clarify.

**Relation To Broader Scientific Literature:**

- This paper focuses on improving the challenging geometric consistency issue in existing dragging-based image editing works, showing an interesting and promising improvement on a specific kind of dragging editing that focuses on rigid transformations.
- The paper proposes VFD-Bench as an addition to existing dragging-based editing benchmark, e.g., DragBench. This VFD benchmark provides ground truths by leveraging video frames. It effectively provides fair evaluation capabilities and could have a positive impact on the field.

**Theoretical Claims:**

NA

---

> ### Author Rebuttal · Authors · 2025-03-31
>
> **[Q1] The method facilitates overall geometric consistency, but edited images still show some fine-grained inconsistencies, e.g., the hat shape in the first sample of Fig.6.**
>
> **[A1]** Yes, we agree with the reviewer’s observation. While FlowDrag significantly improves overall geometric consistency, some fine-grained geometric inconsistencies remain (e.g., the hat shape in Fig. 6). We speculate that this issue primarily arises due to the inherent limitation of the pre-trained 2D Stable Diffusion model (version 1.5), specifically its insufficient 3D understanding. This limitation is commonly observed across diffusion-based drag editing methods. Nonetheless, FlowDrag mitigates overall geometric inconsistency and shows enhanced robustness compared to existing methods (please refer to detailed comparisons in Tables 1 and 2, as well as Figures 6 and 9).
> To further illustrate this strength, we provide additional visualization results in Fig. 10 (additional comparisons on VFD-Bench and Drag100 datasets, including extra examples from Prompt-to-Prompt) and detailed visualizations of our mesh-guided editing process in Fig. 11.
> Additionally, we believe that utilizing backbones inherently capable of stronger geometric reasoning (e.g., video diffusion models) could potentially address and further reduce such fine-grained inconsistencies. We hope our work and results inspires future research in this promising direction.
>
> Fig 10. https://anonymous.4open.science/r/FlowDrag-B950/Fig_10.pdf
>
> Fig 11. https://anonymous.4open.science/r/FlowDrag-B950/Fig_11.pdf
>
>
>
>
> **[Q2] How important and robust is the 3D mesh reconstruction? Additional ablations or insights would help clarify this.**
>
> **[A2]** To analyze the importance and robustness of the 3D mesh reconstruction, we conducted additional sensitivity analyses for both DepthMesh and DiffMesh, as shown in Fig. 13 (please refer to the link provided below).
> For DepthMesh (Fig. 13(a)-(b)), robustness is evaluated by varying the reduction ratio, which directly controls the density of facet connections during mesh construction (as detailed in Supplementary Algorithm 1, Step 3). Specifically, a ratio of 1 indicates a fully connected mesh, while lower ratios substantially reduce vertices and facets, causing geometry degradation and unintended outcomes in mesh deformation and subsequent drag editing. We performed experiments on 20 images from DragBench. Since DragBench lacks ground-truth edited images, we quantified robustness by computing the ratios of metrics relative to the highest-quality mesh (reduction ratio = 1, used as reference image), defined explicitly as follows:
>
> $$ \text{PSNR ratio} = \frac{\text{PSNR (source image)}}{\text{PSNR (reference image)}} $$
>
> $$ \text{1–LPIPS ratio} = \frac{\text{1–LPIPS (source image)}}{\text{1–LPIPS (reference image)}} $$
>
> As shown in Fig. 13(a)-(b), FlowDrag demonstrates stable and robust editing outcomes within an effective reduction ratio range (0.001–1).
> For DiffMesh (Fig. 13(c)-(d)), we similarly assessed robustness by varying the diffusion sampling steps (40, 20, 10, and 5) in the image-to-3D mesh generation process (using Hunyuan3D 2.0). Evaluations on the same 20 DragBench images with identical metrics revealed that sampling steps between 10 and 40 consistently maintained overall object geometry. However, at sampling steps below 10 (e.g., step=5), geometry degraded significantly, causing unintended deformation and editing results. Nevertheless, FlowDrag showed strong robustness for sampling steps of 10 or higher (Fig. 13(c)-(d)).
> These detailed analyses confirm FlowDrag’s robustness across various mesh reconstruction conditions, validating our method’s effectiveness.
>
> Fig 13. https://anonymous.4open.science/r/FlowDrag-B950/Fig_13.pdf

---

### Decision · Program_Chairs · 2025-05-01

**Decision:**

Accept (spotlight poster)

**Comment:**

This paper presents a novel and well-motivated approach to improving geometric consistency in drag-based image editing by introducing a 3D mesh as an intermediate representation. The method effectively addresses the challenge of rigid transformations by leveraging off-the-shelf tools to generate 3D meshes, using energy-function-based deformation techniques (ARAP/SR ARAP) for progressive mesh editing, and translating 3D changes into 2D flow fields for motion supervision. The proposed VFD-Bench dataset addresses a critical gap in existing benchmarks by providing ground-truth editing results, enabling fair and objective evaluation of editing methods. The paper demonstrates promising improvements in geometric consistency and editing quality, supported by both quantitative and qualitative results.

The method is innovative in explicitly incorporating 3D geometric priors into diffusion-based editing, which helps overcome the limitations of relying solely on 2D data. The progressive deformation process and careful pipeline design further enhance the robustness and effectiveness of the approach. While the paper does not compare with DragGAN, a key baseline in this area, and may have limitations in supporting content creation or complex deformations, the contributions to improving geometric consistency and advancing evaluation methodologies are significant. The high-quality qualitative results and the introduction of VFD-Bench strengthen the paper's impact on the field. Overall, the paper provides a solid solution to a challenging problem and makes a meaningful contribution to drag-based image editing.